# Finite-Time Analysis of Entropy-Regularized Neural Natural Actor-Critic Algorithm

**Semih Cayci**                                                                *cayci@mathc.rwth-aachen.de*
*Department of Mathematics*
*RWTH Aachen University*

**Niao He**                                                                          *niao.he@inf.ethz.ch*
*Department of Computer Science*
*ETH Zurich*

**R. Srikant**                                                                        *rsrikant@illinois.edu*
*ECE and CSL*
*University of Illinois at Urbana-Champaign*

**Reviewed on OpenReview:** *https://openreview.net/forum?id=BkEqk7pS1I*

## Abstract

Natural actor-critic (NAC) and its variants, equipped with the representation power of neural networks, have demonstrated impressive empirical success in solving Markov decision problems with large (potentially infinite) state spaces. In this paper, we present a finite-time analysis of NAC with neural network approximation, and identify the roles of neural networks, regularization and optimization techniques (e.g., gradient clipping and weight decay) to achieve provably good performance in terms of sample complexity, iteration complexity and overparametrization bounds for the actor and the critic. In particular, we prove that (i) entropy regularization and weight decay ensure stability by providing sufficient exploration to avoid near-deterministic and strictly suboptimal policies and (ii) regularization leads to sharp sample complexity and network width bounds in the regularized MDPs, yielding a favorable bias-variance tradeoff in policy optimization. In the process, we identify the importance of uniform approximation power of the actor neural network to achieve global optimality in policy optimization due to distributional shift.

## 1 Introduction

In reinforcement learning (RL), an agent aims to find an optimal policy that maximizes the expected total reward in a Markov decision process (MDP) by interacting with an unknown and dynamical environment (Sutton & Barto, 2018; Szepesvári, 2010; Bertsekas & Tsitsiklis, 1996). Policy gradient methods, which employ first-order optimization methods to find the best policy within a parametric policy class, have demonstrated impressive success in numerous complicated RL problems (Williams, 1992; Sutton et al., 1999; Konda & Tsitsiklis, 2000). The success largely benefits from the versatility of policy gradient methods in accommodating a rich class of function approximation schemes as demonstrated by Mnih et al. (2016); Silver et al. (2016); Nachum et al. (2017); Duan et al. (2016).

Natural policy gradient (NPG), natural actor-critic (NAC) and their variants, which use Fisher information matrix as a pre-conditioner for the gradient updates (Amari, 1998; Kakade, 2001; Bhatnagar et al., 2007; Peters & Schaal, 2008), are particularly popular because of their impressive empirical performance in practical applications. In practice, NPG/NAC methods are further combined with (a) neural network approximation for high representation power of both the actor and the critic, and (b) entropy regularization for stability and sufficient exploration, leading to remarkable performance in complicated control tasks that involve large state-action spaces (Haarnoja et al., 2018; Nachum et al., 2017; Ahmed et al., 2019).

Despite the empirical successes, a strong theoretical understanding of policy gradient methods, especially when boosted with function approximation and entropy regularization, appears to be in a nascent stage. Recently, there has been a plethora of theoretical attempts to understand the convergence properties of policy gradient methods and the role of entropy regularization; see, e.g., Agarwal et al. (2020); Bhandari & Russo (2019); Lan (2021); Cen et al. (2020); Mei et al. (2020), just to name a few. These works predominantly study the tabular setting, where a parallelism between the well-known policy iteration and policy gradient methods can be exploited to establish the convergence results. But for the more intriguing function approximation regime, especially with neural network approximation, little theory is known. Two of the main challenges come from the highly nonconvex nature of the problem when using neural network approximation for both the actor and the critic, and the complex exploration dynamics.

In this paper, we provide the first non-asymptotic analysis of an entropy-regularized natural actor-critic (NAC) method in which we use two separate two-layer neural networks for the actor and critic, and employ a learning scheme based on approximate natural policy gradient updates to achieve optimality. We show that the expressive power of these neural networks provide the ability to achieve optimality within a broad class of policies.

## 1.1 Main Contributions

We elaborate some of our contributions below.

- *Sharp sample complexity, convergence rate and overparameterization bounds:* We prove sharp convergence guarantees in terms of sample complexity, iteration complexity and network width. Particularly, we prove that the NAC method with an adaptive step-size achieves sharp $\tilde{O}(1/\epsilon)$ iteration complexity and $\tilde{O}(1/\epsilon^5)$ sample complexity to achieve an $\epsilon$-gap with the optimal policy of the regularized MDP under mildest distribution mismatch conditions to the best of our knowledge. The required network width for both the actor and critic are $\tilde{O}(1/\epsilon^4)$ and $\tilde{O}(1/\epsilon^2)$, respectively. Under the standard distribution mismatch assumption as used by Wang et al. (2019), our sample complexity bound for the unregularized MDP is $\tilde{O}(1/\epsilon^6)$, which improves the existing bounds significantly.

- *Stable policy optimization in the overparameterized regime:* Existing works on neural policy gradient methods with neural network approximation *assume* that the policies perform sufficient exploration to avoid instability, i.e., convergence to near-deterministic and strictly suboptimal stationary policies. In this paper, we prove that policy optimization is stabilized by incorporating (i) overparameterization, (ii) entropy regularization, (iii) gradient clipping, and (iv) weight-decay. In particular, we show that the combination of these methods leads to *"persistence of excitation"* condition, which ensures sufficient exploration to avoid near-deterministic and strictly suboptimal stationary policies. Consequently, we prove convergence to the globally optimal policy under the mildest concentrability coefficient assumption for on-policy NAC to the best of our knowledge.

- *Understanding the dynamics of neural network approximation in policy optimization:* Our analysis reveals that the uniform approximation power of the actor network to approximate Q-functions throughout policy optimization steps is crucial to ensure global (near-)optimality, which is a specific feature of reinforcement learning that induces a distributional shift over time in contrast to a static supervised learning problem. To that end, we establish high-probability bounds for a two-layer feedforward actor neural network to uniformly approximate Q-functions of the policy iterates during the training.

## 1.2 Related Work

*Policy gradient and actor-critic:* Policy gradient methods use a gradient-based scheme to find the optimal policy (Williams, 1992; Sutton et al., 1999). Kakade (2001) proposed the natural gradient method, which uses the Fisher information matrix as a pre-conditioner to fit the problem geometry better. Actor-critic method, which learns approximations to both state-action value functions and policies for variance reduction, was introduced by Konda & Tsitsiklis (2000).

*Neural actor-critic methods:* Recently, there has been a surge of interest in direct policy optimization methods for solving MDPs with large state spaces by exploiting the representation power of deep neural networks. Particularly, deterministic policy gradient (Silver et al., 2014), trust region policy optimization (TRPO) (Schulman et al., 2015), proximal policy optimization (PPO) (Schulman et al., 2017), soft actor-critic (SAC) (Haarnoja et al., 2018; Lee et al., 2020) achieved impressive empirical success in solving complicated control tasks.

*Role of regularization:* Entropy regularization is an essential part of policy optimization algorithms (e.g., TRPO, PPO and SAC) to encourage exploration and achieve fast and stable convergence. It has been numerically observed that entropy regularization leads to a smoother optimization landscape, which leads to improved convergence properties in policy optimization (Ahmed et al., 2019). For tabular reinforcement learning, the impact of entropy regularization was studied by Agarwal et al. (2020); Cen et al. (2020); Mei et al. (2020). On the other hand, the function approximation regime leads to considerably different dynamics compared to the tabular setting mainly because of the generalization over a large state space, complex exploration dynamics and distributional shift. As such, the role of regularization is very different in the function approximation regime, which we study in this paper.

*Theoretical analysis of policy optimization methods:* Despite the vast literature on the practical performance of PG/AC/NAC type algorithms, their theoretical understanding has remained elusive until recently. In the tabular setting, global convergence rates for PG methods were established by Agarwal et al. (2020); Bhandari & Russo (2019); Khodadadian et al. (2021b). By incorporating entropy regularization, it was shown that the convergence rate can be improved significantly in the tabular setting (Shani et al., 2020; Lan, 2021; Cen et al., 2020; Zhan et al., 2021). Recently, finite-time performances of off-policy actor-critic methods in the tabular and linear function approximation regimes were investigated; see e.g., the works by Khodadadian et al. (2021a); Chen et al. (2022). In our paper, we consider neural network approximation under entropy regularization with on-policy sampling.

On the other hand, when the controller employs a function approximator for the purpose of generalization to a large state-action space, the convergence properties of policy optimization methods radically change due to more complicated optimization landscape and distribution mismatch phenomenon in reinforcement learning (Agarwal et al., 2020). Under strong assumptions on the exploratory behavior of policies throughout learning iterations, global optimality of NPG with linear function approximation up to a function approximation error was established by Agarwal et al. (2020). For actor-critic and natural actor-critic methods with linear function approximation, there are finite-time analyses by Chen et al. (2021); Xu et al. (2020); Zhang et al. (2021). For general actor schemes with linear critic, convergence to stationary points was investigated by Kumar et al. (2019); Wu et al. (2020); Qiu et al. (2021).

By incorporating entropy regularization, it was shown that improved convergence rates under much weaker conditions on the underlying controlled Markov chain can be established by Cayci et al. (2021) with linear function approximation. Our paper uses results from the drift analysis in that work, but addresses the significantly challenging complications due to using neural networks with ReLU activation functions in the overparameterized regime, and establishes global convergence to the optimal policies. The neural network approximation eliminates the function approximation error, which is a constant in linear function approximation, by employing a sufficiently wide actor neural network.

*Neural network analysis:* The empirical success of neural networks, which have more parameters than the data points, has been theoretically explained by Jacot et al. (2018); Du et al. (2018); Arora et al. (2019), where it was shown that overparameterized neural networks trained by using first-order optimization methods achieve good generalization properties. The need for massive overparameterization was addressed by Ji et al. (2019); Oymak & Soltanolkotabi (2020), and it was shown that considerably smaller network widths can suffice to achieve good training and generalization results in structured supervised learning problems. Our analysis in this work is mainly inspired by the work of Ji et al. (2019). On the other hand, reinforcement learning problem has significantly different and more challenging dynamics than the supervised learning setting as we have a dynamic optimization problem in actor-critic, where distributional shift occurs as the policies are updated. As such, uniform approximation power of the actor network in approximating various functions through policy optimization steps becomes critical, different from the supervised learning setting

(Ji & Telgarsky, 2019). Our analysis utilizes tools from the works Ji & Telgarsky (2019) and Ji et al. (2019): (i) we consider max-norm geometry to achieve mild overparameterization, (ii) we bound the distance between the neural tangent kernel (NTK) function class and the class of functions realizable by a finite-width neural network by extending the ReLU analysis by Ji et al. (2019); Cayci et al. (2023). The distribution shift in the system due to the dynamical RL setting also yields a significant challenge compared to the previous works, which we address in this paper.

The most relevant work in the literature is Wang et al. (2019), where the convergence of NAC with a two-layer neural network was studied without entropy regularization. It was shown that, under strong assumptions on the exploratory behavior of policies throughout the trajectory, neural-NPG achieves $\epsilon$-optimality with $O(1/\epsilon^{14})$ sample complexity and $O(1/\epsilon^{12})$ network width bounds. In this paper, we incorporate widely-used algorithmic techniques (entropy regularization, weight decay and gradient clipping) to NAC with neural network approximation, and prove significantly improved sample complexity and overparameterization bounds under weaker assumptions on the concentrability coefficients. Additionally, our analysis reveals that the uniform approximation power of the actor neural network is critically important to establish global optimality, where distributional shift plays a crucial role. In another relevant work, Fu et al. (2020) considers a single-timescale actor-critic with neural network approximation, but the function approximation error was not investigated due to the realizability assumption, which assumes that all policies throughout the policy optimization steps are realizable by the neural network. One of the main goals of our work is to study the benefits of employing neural networks in policy optimization, and we explicitly characterize the function class and approximation error that stems from the use of finite-width neural networks.

## 1.3 Notation

For a sequence of numbers $\{x_i : i \in I\}$ where $I$ is an index set, $[x_i]_{i \in I}$ denotes the vector obtained by concatenation of $x_i, i \in I$. For a set $A$, $|A|$ denotes its cardinality. For two distributions $P, Q$ defined over the same probability space, Kullback-Leibler divergence is denoted as follows: $D_{KL}(P\|Q) = \mathbb{E}_{s \sim P}\left[\log \frac{P(s)}{Q(s)}\right]$. For a convex set $C \subset \mathbb{R}^d$ and $x \in \mathbb{R}^d$, $\mathcal{P}_C(x)$ denotes the projection of $x$ onto $C$: $\mathcal{P}_C(x) = \arg\min_{y \in C} \|x - y\|_2$. For $n \in \mathbb{Z}^+$, $[n] = \{1, 2, \ldots, n\}$. For $d, m \in \mathbb{N}$, $R > 0$ and $v \in \mathbb{R}^{m \times d}$, we denote

$$\mathcal{B}_{m,R}^d(v) = \left\{y \in \mathbb{R}^{m \times d} : \sup_{i \in [m]} \|v_i - y_i\|_2 \leq \frac{R}{\sqrt{m}}\right\},$$

where $v_i$ denotes the $i^{\text{th}}$ row of $v$. $\mathbb{1}_A$ denotes the indicator function for any event $A$.

## 2 Background and Problem Setting

In this section, we introduce basic backgrounds of the problem setting, the natural actor critic method, as well as entropy regularization and neural network approximation that we consider.

### 2.1 Markov Decision Processes

We consider a discounted Markov decision process $(\mathcal{S}, \mathcal{A}, P, r, \gamma)$ where $\mathcal{S}$ and $\mathcal{A}$ are the state and action spaces, $P$ is a (unknown) transition kernel, $r : \mathcal{S} \times \mathcal{A} \to [0, r_{max}]$, $0 < r_{max} < \infty$ is the reward function, and $\gamma \in (0, 1)$ is the discount factor. In this work, we consider a state space $\mathcal{S}$ and a finite action space $\mathcal{A}$ such that $\mathcal{S} \times \mathcal{A} \subset \mathbb{R}^d$. Also, we assume that, by appropriate representation of the state and action variables, the following bound holds: $\|(s, a)\|_2 \leq 1$, throughout the paper.

**Value function:** Under a randomized *policy* $\pi : \mathcal{S} \to \mathcal{A}$, an action $a \in \mathcal{A}$ is taken at a given state $s \in \mathcal{S}$ with probability $\pi(a|s)$. A policy $\pi$ introduces a trajectory by specifying $a_t \sim \pi(\cdot|s_t)$ and $s_{t+1} \sim P(\cdot|s_t, a_t)$. For any $s_0 \in \mathcal{S}$, the corresponding value function of a policy $\pi$ is as follows:

$$V^\pi(s_0) = \mathbb{E}\Big[\sum_{t=0}^\infty \gamma^t r(s_t, a_t)|s_0\Big], \tag{1}$$

where $a_t \sim \pi(\cdot|s_t)$ and $s_{t+1} \sim P(\cdot|s_t, a_t)$.

**Entropy regularization:** In order to avoid near-deterministic suboptimal policies in policy optimization by encouraging exploration, entropy regularization is commonly used in practice (Silver et al., 2016; Haarnoja et al., 2018; Nachum et al., 2017; Ahmed et al., 2019). For a policy $\pi$, let

$$H^\pi(s_0) = \mathbb{E}\Big[\sum_{t=0}^{\infty} \gamma^t \mathcal{H}\big(\pi(\cdot|s_t)\big)\Big|s_0\Big], \tag{2}$$

where $\mathcal{H}(\pi(\cdot|s)) = -\sum_{a \in \mathcal{A}} \pi(a|s) \log\big(\pi(a|s)\big)$ is the entropy functional. Then, for the regularization parameter $\lambda > 0$, the entropy-regularized value function is defined as follows:

$$V_\lambda^\pi(s_0) = V^\pi(s_0) + \lambda H^\pi(s_0). \tag{3}$$

The max-entropy policy, which assigns probability $1/|\mathcal{A}|$ to each action in each state, maximizes the regularizer $H^\pi(s_0)$ for any $s_0 \in \mathcal{S}$. Thus, the regularizer $H^\pi(s_0)$ term in equation 3 encourages exploration, controlled by $\lambda > 0$.

**Entropy-regularized objective:** For a given initial state distribution $\mu$ and for a given regularization parameter $\lambda > 0$, the objective in this paper is to maximize the entropy-regularized value function

$$\max_\pi V_\lambda^\pi(\mu), \tag{4}$$

where $V_\lambda^\pi(\mu) := \mathbb{E}_{s_0 \sim \mu}[V_\lambda^\pi(s_0)]$. We denote the optimal policy for the regularized MDP as $\pi^*$ throughout the paper.

**Q-function and advantage function:** The (entropy-regularized) Q-function $q_\lambda^\pi(s, a)$ is defined as:

$$q_\lambda^\pi(s, a) = \mathbb{E}\Big[\sum_{k=0}^{\infty} \gamma^k \big(r(s_k, a_k) - \lambda \log \pi(a_k|s_k)\big)\Big|s_0 = s, a_0 = a\Big]. \tag{5}$$

Note that $q_\lambda^\pi$ is the fixed point of the Bellman equation $q(s, a) = \mathcal{T}^\pi q(s, a)$ where the Bellman operator $\mathcal{T}^\pi$ is defined as:

$$\mathcal{T}^\pi q(s, a) = r(s, a) - \lambda \log \pi(a|s) + \gamma \mathbb{E}_{s' \sim P(\cdot|s,a), a' \sim \pi(\cdot|s')}[q(s', a')]. \tag{6}$$

As we will see, for NAC algorithms, the following function, called the soft Q-function under a policy $\pi$, turns out to be a useful quantity (Cen et al., 2020):

$$Q_\lambda^\pi(s, a) = r(s, a) + \gamma \mathbb{E}_{s' \sim P(\cdot|s,a)}[V_\lambda^\pi(s')]. \tag{7}$$

Note that the two Q-functions are related as follows:

$$q_\lambda^\pi(s, a) = Q_\lambda^\pi(s, a) - \lambda \log \pi(a|s).$$

The advantage function under a policy $\pi$ is defined as follows:

$$A_\lambda^\pi(s, a) = q_\lambda^\pi(s, a) - V_\lambda^\pi(s). \tag{8}$$

Similarly, the soft advantage function is defined as follows:

$$\Xi_\lambda^\pi(s, a) = Q_\lambda^\pi(s, a) - \sum_{a' \in \mathcal{A}} \pi(a'|s) Q_\lambda^\pi(s, a'). \tag{9}$$

Lastly, we can bound the entropy-regularized value function as follows:

$$0 \le V_\lambda^\pi(\mu) \le \frac{r_{max} + \lambda \log |\mathcal{A}|}{1 - \gamma}, \tag{10}$$

for any $\lambda > 0, \pi$ since $r \in [0, r_{max}]$ and $\mathcal{H}(p) \le \log |\mathcal{A}|$ for any distribution $p$ over $\mathcal{A}$ (Cen et al., 2020).

## 2.2 Natural Policy Gradient under Entropy Regularization

For a given randomized policy $\pi_\theta$ parameterized by $\theta \in \Theta$ where $\Theta$ is a given parameter space, policy gradient methods maximize $V^{\pi_\theta}(\mu)$ by using the policy gradient $\nabla_\theta V^{\pi_\theta}(\mu)$. Natural policy gradient, as a quasi-Newton method, adjusts the gradient update to fit problem geometry by using the Fisher information matrix as a pre-conditioner (Kakade, 2001; Cen et al., 2020).

Let

$$G^{\pi_\theta}(\mu) = \mathbb{E}_{s \sim d_\mu^{\pi_\theta}, a \sim \pi_\theta(\cdot|s)}\Big[\nabla_\theta \log \pi_\theta(a|s)\nabla_\theta^\top \log \pi_\theta(a|s)\Big],$$

be the Fisher information matrix under policy $\pi_\theta$, where $d_\mu^\pi(\cdot) = (1-\gamma)\sum_{k=0}^\infty \gamma^k \mathbb{P}(s_k \in \cdot|s_0 \sim \mu)$, is the discounted state visitation distribution under a policy $\pi$. Then, the update rule under NPG can be expressed as

$$\theta \leftarrow \theta + \eta \cdot \big[G^{\pi_\theta}(\mu)\big]^{-1}\nabla V_\lambda^{\pi_\theta}(\mu), \tag{11}$$

where $\eta > 0$ is the step-size. Equivalently, the NPG update can be written as follows:

$$\theta^+ \in \arg\max_{\theta \in \mathbb{R}^d}\Big\{\nabla_\theta^\top V_\lambda(\pi_{\theta^-})(\theta - \theta^-) - \frac{1}{2\eta}(\theta - \theta^-)^\top G^{\pi_{\theta^-}}(\mu)(\theta - \theta^-)\Big\}. \tag{12}$$

The above update scheme is closely related to gradient ascent and policy mirror ascent. Note that the gradient ascent for policy optimization performs the following update:

$$\theta^+ \in \arg\max_{\theta \in \mathbb{R}^d}\Big\{\nabla_\theta^\top V_\lambda(\pi_{\theta^-})(\theta - \theta^-) - \frac{1}{2\eta}\|\theta - \theta^-\|_2^2\Big\}. \tag{13}$$

The update in equation 13 leads to the policy gradient algorithm (Williams, 1992). Compared to equation 13, the natural policy gradient uses a generalized Mahalanobis distance (i.e., weighted-$\ell_2$ distance) as the Bregman divergence instead of $\ell_2$ distance (Cen et al., 2020; Lan, 2021; Agarwal et al., 2020).

In the following, we provide necessary tools to compute the policy gradient and the update rule in equation 11 based on Cayci et al. (2021).

**Proposition 1** (Policy gradient). *For any $\theta$ and $\lambda > 0$, we have:*

$$\nabla_\theta V_\lambda^{\pi_\theta}(\mu) = \frac{1}{1-\gamma}\mathbb{E}_{s \sim d_\mu^{\pi_\theta}, a \sim \pi_\theta(\cdot|s)}\Big[\nabla_\theta \log \pi_\theta(a|s)q_\lambda^{\pi_\theta}(s,a)\Big]. \tag{14}$$

Based on Proposition 1, the gradient update of natural policy gradient can be computed by the following lemma, which is an extension of the works by Kakade (2001); Agarwal et al. (2020); Cayci et al. (2021).

**Lemma 1.** *Let*

$$L(w,\theta) = \mathbb{E}_{s \sim d_\mu^{\pi_\theta}, a \sim \pi_\theta(\cdot|s)}\Big[\big(\nabla_\theta^\top \log \pi_\theta(a|s)w - q_\lambda^{\pi_\theta}(s,a)\big)^2\Big], \tag{15}$$

*be the error for a given policy parameter $\theta$. Define*

$$w_\lambda^{\pi_\theta} \in \arg\min_w L(w,\theta). \tag{16}$$

*Then, we have:*

$$G^{\pi_\theta}(\mu)w_\lambda^{\pi_\theta} = (1-\gamma)\nabla_\theta V_\lambda^{\pi_\theta}(\mu), \tag{17}$$

*where $G^{\pi_\theta}$ is the Fisher information matrix.*

The above results for general policy parameterization will provide basis for the entropy-regularized natural actor-critic (NAC) with neural network approximation that we will introduce in the following section, with certain modifications for variance reduction and stability that we will describe; see Remark 2 later.

# 3 Natural Actor-Critic with Neural Network Approximation

In this section, we introduce the entropy-regularized natural actor critic algorithm formally, where both the actor and critic are represented by single-hidden-layer neural networks.

Throughout this paper, we make the following assumption on the sampling process, which is standard in policy optimization (Agarwal et al., 2020).

**Assumption 1** (Sampling oracle). *For a given initial state distribution $\mu$ and policy $\pi$, we assume that the controller is able to obtain an independent sample from $d_\mu^\pi$ at any time.*

The sampling process involves a resetting mechanism and a simulator, which are available in many important application scenarios, and sampling from a state visitation distribution $d_\mu^\pi$ can be performed by using the sampler in Algorithm 4 without the knowledge of $d_\mu^\pi$ (Agarwal et al., 2020; Konda & Tsitsiklis, 2003). For further discussion on the sampling and its impacts on the analysis, please see Section B.

## 3.1 Actor Network and Natural Policy Gradient

For a network width $m \in \mathbb{Z}^+$ and $c_i \in \mathbb{R}$, $\theta_i \in \mathbb{R}^d$ for $i \in [m]$, the actor network is given by the single-hidden-layer neural network:

$$f(s, a; (c, \theta)) = \frac{1}{\sqrt{m}} \sum_{i=1}^{m} c_i \sigma\big(\langle \theta_i, (s, a) \rangle\big), \tag{18}$$

where $c = [c_i]_{i \in [m]}, \theta = [\theta_i]_{i \in [m]}$, $\sigma(x) = \max\{0, x\}$ is the ReLU activation function. As a common practice (Ji et al., 2019; Oymak & Soltanolkotabi, 2020; Arora et al., 2019), we fix the output layer $c$ after a random initialization, and only train the weights of hidden layer, namely, $\theta \in \Theta \subset \mathbb{R}^{m \times d}$. We note that training the output layer $c$ would lead to an additional additive kernel term in the neural tangent kernel, resulting in a function class that is at least as rich as the original one (Cucker & Zhou, 2007). We consider a fixed output layer in this work for the sake of simplicity.

Given a (possibly random) parameter $\theta^0 \in \mathbb{R}^{m \times d}$, a design parameter $R > 0$, regularization parameter $\lambda > 0$ and network width $m \in \mathbb{Z}^+$, the parameter space that we consider is as follows:

$$\Theta = \left\{ \theta \in \mathbb{R}^{m \times d} : \max_{i \in [m]} \|\theta_i - \theta_i^0\|_2 \leq \frac{R}{\lambda \sqrt{m}} \right\}. \tag{19}$$

For this parameter space $\Theta$, the policy class that we consider is $\Pi = \{\pi_\theta : \theta \in \Theta\}$, where the policy that corresponds to $\theta \in \Theta$ is as follows:

$$\pi_\theta(a|s) = \frac{\exp(f(s, a; (c, \theta)))}{\sum_{a' \in \mathcal{A}} \exp(f(s, a'; (c, \theta)))}. \tag{20}$$

We randomly initialize the actor neural network by using the symmetric initialization in Algorithm 1, $(c, \theta(0)) \sim \texttt{sym\_init}(m, d)$ (Bai & Lee, 2019). Later, we will employ a similar symmetric initialization scheme for the critic neural network.

---

**Algorithm 1:** $\texttt{sym\_init}(m, d)$ - Symmetric Initialization

**inputs:** $m$: network width, $d$: ambient dimension;
**for** $i = 1, 2, \ldots, m/2$ **do**
  $c_i = -c_{i+m/2} \sim Rademacher$;
  $\theta_i = \theta_{i+m/2} \sim \mathcal{N}(0, I_d)$;
**return** network weights $(c, \theta)$

---

We denote the policy at iteration $t < T$ as $\pi_t = \pi_{\theta(t)}$ and neural network output as $f_t(s, a) = f(s, a; (c, \theta(t)))$. In the absence of the prior knowledge of $\Xi_\lambda^{\pi_t}$ and $d_\mu^{\pi_t}$, we construct a stochastic estimate of

$$u_t^\star = \min_{u \in \mathcal{B}_{m,R}^d(0)} \mathbb{E}_{s \sim d_\mu^{\pi_t}, a \sim \pi_t(\cdot|s)} [(\nabla^\top \log \pi_t(a|s) u - \Xi_\lambda^{\pi_t}(s, a))^2], \tag{21}$$

by using samples from the system via the following actor-critic meta-algorithm:

- **Critic:** Temporal difference learning algorithm (Algorithm 3 in Section 3.2), which employs a critic neural network, returns a set of neural network weights that yield a sample-based estimate for the soft advantage function $\{\widehat{\Xi}_\lambda^{\pi_t}(s,a) : (s,a) \in \mathcal{S} \times \mathcal{A}\}$.

- **Policy update:** Given this, we construct a stochastic estimate of $u_t^\star$ by using stochastic gradient descent (SGD) with $N$ iterations and step-size $\alpha_A > 0$. To that end, starting with $u_0^{(t)} = 0$, an iteration of SGD is as follows

$$u_{n+1/2}^{(t)} = u_n^{(t)} - \alpha_A \Big( \nabla_\theta^\top \log \pi_t(a_n|s_n) u_n^{(t)} - \widehat{\Xi}_\lambda^{\pi_t}(s_n, a_n) \Big) \nabla_\theta \log \pi_t(a_n|s_n), \tag{22}$$

$$u_{n+1}^{(t)} = \mathcal{P}_{\mathcal{B}_{m,R}^d(0)} \Big( u_{n+1/2}^{(t)} \Big), \tag{23}$$

where $s_n \sim d_\mu^{\pi_t}$ and $a_n \sim \pi_t(\cdot|s_n)$ for $n = 0, 1, \ldots, N-1$, $\widehat{\Xi}_\lambda^{\pi_t}$ is the output of the critic. Then, the final estimate is $u_t = \frac{1}{N} \sum_{n=1}^N u_n^{(t)}$. By using $u_t$, we perform the following update:

$$\theta(t+1) = \theta(t) + \eta_t \cdot w_t,$$

where $w_t = u_t - \lambda\big(\theta(t) - \theta(0)\big)$.

The natural actor-critic algorithm is summarized as a meta-algorithm in Algorithm 2. Below, we summarize the modifications in the algorithm that we consider in this paper with respect to the NPG described in the previous section.

**Remark 1** (Weight decay and projection)**.** The update in each iteration of the NAC algorithm described in Algorithm 2 can be equivalently written as follows:

$$\theta(t+1) - \theta(0) = (1 - \eta_t\lambda) \cdot (\theta(t) - \theta(0)) + (1 - \eta_t)u_t, \tag{24}$$

where $u_t$ is an approximate solution to the optimization problem (21). As we will see, the projection of $u_t$ onto $\mathcal{B}_{m,R}^d(0)$ (which can be considered as gradient clipping since this operation clips the natural policy gradient update $u_t$ to ensure $\max_{1 \le i \le m} \|u_{i,t}\|_2 \le R/\sqrt{m}$), in conjunction with the weight decay in the policy update (24) enables us to control $\max_{i \in [m]} \|\theta_i(t) - \theta_i(0)\|_2$ while taking (natural) gradient steps towards the optimal policy. Controlling $\max_{i \in [m]} \|\theta_i(t) - \theta_i(0)\|_2$ is critical for two reasons: (i) to ensure sufficient exploration to achieve global optimality (see Proposition 2), and (ii) to establish the so-called kernel regime, which holds near the random initialization (see Lemma 3 and Theorem 1).

Alternatively, one may be tempted to project $\theta(t)$ onto a ball around $\theta(0)$ in the $\ell_2$-geometry to control $\max_i \|\theta_i(t) - \theta_i(0)\|_2$. However, as the algorithm follows the *natural* policy gradient, which uses a different Bregman divergence than $\|\cdot\|_2$, projection of $\theta(t)$ with respect to the $\ell_2$-norm may not result in moving the policy in the direction of improvement. Similarly, since we parameterize the policies by using a lower-dimensional vector $\theta \in \mathbb{R}^{m \times d}$ to avoid storing and computing $|\mathcal{S} \times \mathcal{A}|$-dimensional policies, Bregman projection in the probability simplex, which is commonly used in direct parameterization, is not a feasible option for policy optimization with function approximation. As such, simultaneous use of weight decay and projection of the update $u_t$ are critical to control the network weights and policy improvement.

**Remark 2** (Baseline)**.** Note that the update $u_t^\star$ in equation 21 uses the soft-advantage function $\Xi_\lambda^\pi$ rather than the state-action value function $q_\lambda^\pi$. The soft-advantage function uses $\sum_{a \sim \pi_t} \pi_t(a|s) Q_\lambda^{\pi_t}(s,a)$ as a baseline for variance reduction, which is a common practice in policy gradient methods (Sutton & Barto, 2018).

In the following subsection, we describe the critic algorithm in detail.

## 3.2 Critic Network and Temporal Difference Learning

We estimate $\Xi_\lambda^{\pi_t}$ by using the neural TD learning algorithm with max-norm regularization (Cayci et al., 2023). Note that $\Xi_\lambda^{\pi_t}$ can be directly obtained from $q_\lambda^{\pi_\theta}$ via $Q_\lambda^{\pi_\theta}(s,a) = q_\lambda^{\pi_\theta}(s,a) - \lambda \log \pi_\theta(a|s)$ and equation 9.

---

**Algorithm 2:** Entropy-regularized Neural NAC

---

initialize $(c, \theta(0)) \sim \texttt{sym\_init}(m, d)$;
**for** $t = 0, 1, \ldots, T - 1$ **do**

$\quad$ Critic: $\widehat{\Xi}_\lambda^{\pi_t} = \texttt{MN-NTD}(\pi_t, R, m', T', \alpha_C)$ // See Alg. 3 for details;

$\quad$ Initialize: $u_0^{(t)} = 0$;

$\quad$ **for** $n = 0, 1, \ldots, N$ **do**

$\quad\quad$ Sampling: $s_n \sim d_\mu^{\pi_t}, a_n \sim \pi_t(\cdot|s_n).$;

$\quad\quad$ $u_{n+1/2}^{(t)} = u_n^{(t)} - \alpha_A \left( \nabla_\theta^\top \log \pi_t(a_n|s_n) u_n^{(t)} - \widehat{\Xi}_\lambda^{\pi_t}(s_n, a_n) \right) \nabla_\theta \log \pi_t(a_n|s_n)$;

$\quad\quad$ $u_{n+1}^{(t)} = \mathcal{P}_{\mathcal{B}_{m,R}^d(0)} \left( u_{n+1/2}^{(t)} \right)$;

$\quad$ $u_t = \frac{1}{N} \sum_{n=1}^N u_n^{(t)}$;

$\quad$ $\theta(t + 1) = \theta(t) + \eta_t u_t - \eta_t \lambda[\theta(t) - \theta(0)]$;

---

Since $q_\lambda^{\pi_\theta}$ is the fixed point of the Bellman equation (5), it can be approximated by using temporal difference (TD) learning algorithms.

For the critic, we use a two-layer neural network of width $m'$, which is defined as follows:

$$\widehat{q}(s, a; (b, W)) = \frac{1}{\sqrt{m'}} \sum_{i=1}^{m'} b_i \sigma \left( \langle W_i, (s, a) \rangle \right). \tag{25}$$

The critic network is initialized according to the symmetric initialization scheme in Algorithm 1. Let $(b, W(0))$ denote the initialization.

To estimate the value function, we aim to solve the following problem:

$$W^\star = \arg\min_W \quad \mathbb{E}_{s \sim d_\mu^{\pi_\theta}, a \sim \pi_\theta(\cdot|s)} \left[ \left( \widehat{q}(s, a; (b, W)) - \mathcal{T}^{\pi_\theta} \widehat{q}(s, a; (b, W)) \right)^2 \right]. \tag{26}$$

where $\mathcal{T}^\pi$ is the Bellman operator in equation 6. Particularly, we consider max-norm regularization in the updates of the critic, which was shown to be effective in supervised learning and reinforcement learning (see Goodfellow et al. (2016; 2013); Srivastava et al. (2014); Cayci et al. (2023)). For a given $w_0 \in \mathbb{R}^d$ and $R > 0$, let

$$\mathcal{G}_R(w_0) = \{w \in \mathbb{R}^d : \|w - w_0\|_2 \leq R/\sqrt{m'}\}. \tag{27}$$

Under max-norm regularization, each hidden unit's weight vector is confined within the set $\mathcal{G}_R(W_i(0))$ for a given projection radius $R$.

The critic update for a given policy $\pi_\theta, \theta \in \Theta$ is summarized in Algorithm 3, which we call $\texttt{MN-NTD}$. At each iteration of $\texttt{MN-NTD}$, we perform the update:

$$W_i(k + 1) = \mathcal{P}_{\mathcal{G}_R(W_i(0))} \left( W_i(k) + \alpha \big( r_k + \gamma \widehat{q}_k(s_k', a_k') - \widehat{q}_k(s_k, a_k) \big) \nabla_{W_i} \widehat{q}_k(s_k, a_k) \right), \forall i \in [m'],$$

where $\widehat{q}_k(s, a) = \widehat{q}(s, a; (b, W(k)))$, $r_k = r(s_k, a_k) - \lambda \log \pi_\theta(a_k|s_k)$ and $\mathcal{P}_\mathcal{C}$ is the projection operator onto a set $\mathcal{C} \subset \mathbb{R}^d$. For $k = 0, 1, \ldots, T' - 1$, we assume that $(s_k, a_k)$ is sampled from $d_\mu^{\pi_\theta}$, i.e., $s_k \sim d_\mu^{\pi_\theta}, a_k \sim \pi_\theta(\cdot|s_k)$. Upon obtaining $(s_k, a_k)$, the next state-action pair is obtained by following $\pi_\theta$: $s_k' \sim P(\cdot|s_k, a_k)$, $a_k' \sim \pi_\theta(\cdot|s_k')$. One can replace the i.i.d. sampling here with Markovian sampling at the cost of a more complicated analysis as shown by Bhandari et al. (2021). However, since experience replay is used in practice, the actual sampling procedure is neither purely Markovian or i.i.d., and here for simplicity of the analysis, we choose to model it as i.i.d. sampling.

The output of the critic, which approximates $q_\lambda^{\pi_\theta}$, is then obtained as:

$$\overline{q}_{T'}^{\pi_\theta}(s, a) = \widehat{q}\Big(s, a; \big(b, \frac{1}{T'} \sum_{k<T'} W(k)\big)\Big), \quad (s, a) \in \mathcal{S} \times \mathcal{A},$$

---

**Algorithm 3:** `MN-NTD` - Max-Norm Regularized Neural TD Learning

---

**Inputs:** Policy $\pi_\theta$, proj. radius $R$, network width $m'$, sample size $T'$, step-size $\alpha_C$;
**Initialization:** $(b, W(0)) = \texttt{sym\_init}(m', d)$;
**for** $k < T' - 1$ **do**

    Observe $(s_k, a_k) \sim d_\mu^{\pi_\theta} \circ \pi_\theta(\cdot|s_k), s_k' \sim P(\cdot|s_k, a_k) \ a_k' \sim \pi_\theta(\cdot|s_k')$;

    Observe reward: $r_k := r(s_k, a_k) - \lambda \log \pi_\theta(a_k|s_k)$;

    Compute semi-gradient: $g_k = \left( r_k + \gamma \hat{q}_k(s_k', a_k') - \hat{q}_k(s_k, a_k) \right) \nabla_\theta \hat{q}_k(s_k, a_k)$;

    Take a semi-gradient step: $W(k + 1/2) = W(k) + \alpha_C g_k$;

    Max-norm regularization: $W_i(k + 1) = \mathcal{P}_{\mathcal{G}_R(W_i(0))} \{ W_i(k + 1/2) \}, \forall i \in [m']$;

**return** $\overline{q}_{T'}^{\pi_\theta}(s, a) = \hat{q}\left( s, a; \left( b, \frac{1}{T'} \sum_{k < T'} W(k) \right) \right)$ for all $(s, a) \in \mathcal{S} \times \mathcal{A}$

---

where $T'$ is the number of iterations of `MN-NTD`. We obtain an approximation of the soft Q-function as

$$\overline{Q}_\lambda^{\pi_\theta}(s, a) = \overline{q}_{T'}^{\pi_\theta}(s, a) - \lambda \log \pi_\theta(a|s).$$

The corresponding estimate for the soft advantage function is the following:

$$\widehat{\Xi}_\lambda^{\pi_\theta}(s, a) = \overline{Q}_\lambda^{\pi_\theta}(s, a) - \sum_{a' \in \mathcal{A}} \pi_\theta(a'|s) \overline{Q}_\lambda^{\pi_\theta}(s, a'). \tag{28}$$

## 4 Sample Complexity and Overparameterization Bounds for Neural NAC

In this section, we analyze the convergence of the entropy-regularized neural NAC algorithm and provide sample complexity and overparameterization bounds for both the actor and the critic.

### 4.1 Regularization and Persistence of Excitation under Neural NAC

The persistence of excitation, which broadly refers to a strictly positive exploration probability of each action at each state, is an important general condition to guarantee the convergence in stochastic control problems (Kumar & Varaiya, 2015). The following important proposition states that the persistence of excitation condition is satisfied by the entropy-regularized Neural NAC, which implies sufficient exploration is achieved to ensure convergence to global optimality.

**Proposition 2** (Persistence of excitation under overparameterization)**.** *For any regularization parameter $\lambda > 0$, projection radius $R$, the entropy-regularized NAC satisfies the following:*

$$\max_{i \in [m]} \|\theta_i(t) - \theta_i(0)\|_2 \leq \frac{R \varkappa_t}{\lambda \sqrt{m}}, \tag{29}$$

*where*

$$\varkappa_t = \begin{cases} 1, & \eta_t = \frac{1}{\lambda(t+1)}, \\ 1 - (1 - \eta\lambda)^t, & \eta_t = \eta \in \left(0, \frac{1}{\lambda}\right), \end{cases} \tag{30}$$

*for all $t \geq 0$ almost surely. Consequently,*

$$\pi_{min} := \inf_{(s,a) \in \mathcal{S} \times \mathcal{A}} \pi_t(a|s) \geq \frac{\exp\left( -2R/\lambda - 2\rho_0 \left( \frac{R\varkappa_t}{\lambda}, m, \delta \right) \right)}{|\mathcal{A}|} > 0, \tag{31}$$

*for all $t \geq 0$ with probability at least $1 - \delta$ over the random initialization of the actor network, where the function $\rho_0$ is given by*

$$\rho_0(R_0, m, \delta) = \frac{16 R_0}{\sqrt{m}} \left( R_0 + \sqrt{\log\left(\frac{1}{\delta}\right)} + \sqrt{d \log(m)} \right). \tag{32}$$

Proposition 2 has two critical implications:

(i) Any action $a \in \mathcal{A}$ is explored with strictly positive probability at any given state $s \in \mathcal{S}$, which implies that all policies throughout the policy optimization steps satisfy the "*persistence of excitation*" condition with high probability over the random initialization. As we will see in the convergence analysis, this property implies sufficient exploration, which ensures that near-deterministic suboptimal policies are avoided. Sufficient exploration is achieved by the combination of (i) entropy regularization, (ii) weight decay, (iii) max-norm projection of $u_t$, and (iv) large network width $m$ for the actor network.

(ii) We can control the deviation of the actor network weights by $R$, $\lambda$ and $m$. This property is key for the neural network analysis in the lazy-training regime.

## 4.2 Transportation Mappings and Function Classes

We first present a brief discussion on kernel approximations of neural networks, which will be useful to state our convergence results. Consider the following space of mappings:

$$\mathcal{H}_{\bar{\nu}} = \{v : \mathbb{R}^d \to \mathbb{R}^d : \sup_{w \in \mathbb{R}^d} \|v(w)\|_2 \le \bar{\nu}\}, \tag{33}$$

and the function class:

$$\mathcal{F}_{\bar{\nu}} = \left\{ g(\cdot) = \mathbb{E}_{w_0 \sim \mathcal{N}(0, I_d)}[\langle v(w_0), \cdot \rangle \mathbb{1}_{\{\langle w_0, \cdot \rangle > 0\}}] : v \in \mathcal{H}_{\bar{\nu}} \right\}. \tag{34}$$

Note that $\mathcal{F}_{\bar{\nu}}$ is a provably rich subset of the reproducible kernel Hilbert space (RKHS) induced by the neural tangent kernel, which is dense in the space of continuous functions on a compact set as shown by Ji et al. (2019). For a given class of transportation maps $\mathcal{V} = \{v_k \in \mathcal{H}_{\bar{\nu}} : k \in [K]\}$ for $K \ge 1$, we also consider the following space of functions:

$$\mathcal{F}_{K,\bar{\nu},\mathcal{V}} = \left\{ g(\cdot) = \mathbb{E}_{w_0 \sim \mathcal{N}(0, I_d)}[\langle \sum_{k \in [K]} \alpha_k v_k(w_0), \cdot \rangle \mathbb{1}_{\{\langle w_0, \cdot \rangle > 0\}}] : \|\alpha\|_1 \le 1 \right\}. \tag{35}$$

Note that the above set depends on the choice of $\{v_k\}_{k \in [K]}$ but these maps can be arbitrary. Any separable subspace of continuous functions $f : \mathbb{R}^d \to \mathbb{R}$ over a compact domain has a countable basis $\{\varphi_k : k = 0, 1, \ldots\}$ (Kreyszig, 1991). There exist $v_k \in \mathcal{H}_{\bar{\nu}}$ such that $g_k(s, a) = \mathbb{E}[v_k(w_0) \cdot (s, a) \mathbb{1}_{\{(s,a) \cdot w_0 \ge 0\}}]$ approximates $\varphi_k$ well by the universal approximation results (Ji et al., 2019). As such, $\mathcal{F}_{K,\bar{\nu},\mathcal{V}}$ is able to approximate any separable subspace of continuous functions over a compact space as $K \to \infty$ for sufficiently large $\bar{\nu}$ and an appropriate $\mathcal{V}$.

## 4.3 Convergence of the Critic

We make the following realizability assumption for the Q-function.

**Assumption 2** (Realizability of the Q-function). *For any $t \ge 0$, we assume that $q_\lambda^{\pi_t} \in \mathcal{F}_{\bar{\nu}}$ for some $\bar{\nu} > 0$.*

Assumption 2 is a condition on the class of realizable functions that can be approximated by the critic network, which is dense in the space of continuous functions over $\Omega_d$ (see Section 4.2). One can also replace the above condition by a slightly stronger condition which states that $q_\lambda^{\pi_\theta} \in \mathcal{F}_{\bar{\nu}}$, $\forall \theta \in \Theta$. Note that the class of functions $\mathcal{F}_{\bar{\nu}}$ is deterministic and its approximation properties are well-known (Ji et al., 2019). Wang et al. (2019) assumed that the state-action value functions lie in a *random* function class, which is obtained by shifting $\mathcal{F}_{\bar{\nu}}$ with a Gaussian process. By employing a symmetric initialization, we eliminate this Gaussian process noise, and therefore the realizable class of functions is deterministic and provably rich.

**Lemma 2** (Convergence of the Critic, Theorem 2, Cayci et al. (2023)). *Under Assumption 2, for any error probability $\delta \in (0, 1)$, let*

$$\ell(m', \delta) = 4\sqrt{\log(2m' + 1)} + 4\sqrt{\log(T/\delta)},$$

*and $R > \bar{\nu}$. Then, for any target error $\varepsilon > 0$, number of iterations $T' \in \mathbb{N}$, network width*

$$m' > \frac{16\Big(\bar{\nu} + \big(R + \ell(m', \delta)\big)\big(\bar{\nu} + R\big)\Big)^2}{(1-\gamma)^2 \varepsilon^2},$$

*and step-size*

$$\alpha_C = \frac{\varepsilon^2(1-\gamma)}{(1+2R)^2},$$

*the critic yields the following bound:*

$$\mathbb{E}\left[\sqrt{\mathbb{E}_{s \sim d_\mu^{\pi_t}, a \sim \pi_t(\cdot|s)}\left[\big(\bar{q}_{T'}^{\pi_t}(s,a) - q_\lambda^{\pi_t}(s,a)\big)^2\right]} \mathbb{1}_{A_2}\right] \leq \frac{(1+2R)\bar{\nu}}{\varepsilon(1-\gamma)\sqrt{T'}} + 3\varepsilon,$$

*where $A_2$ holds with probability at least $1 - \delta$ over the random initializations of the critic network.*

Note that in order to achieve a target error less than $\varepsilon > 0$, a network width of $m' = \widetilde{O}\left(\frac{\bar{\nu}^4}{\varepsilon^2}\right)$ and iteration complexity $T' = O\left(\frac{(1+2\bar{\nu})^2\bar{\nu}^2}{\varepsilon^4}\right)$ suffice. The analysis of TD learning algorithm by Cayci et al. (2023) uses results from Ji & Telgarsky (2019), which was given for classification (supervised learning) problems with logistic loss. On the other hand, TD learning requires a significantly more challenging analysis because of bootstrapping in the updates (i.e., using a stochastic semi-gradient instead of a true gradient) and quadratic loss function. Furthermore, for improved sample complexity and overparameterization bounds, max-norm regularization is employed instead of early stopping (Cayci et al., 2023).

## 4.4 Global Optimality and Convergence of Neural NAC

In this section, we provide the main convergence result for the entropy-regularized NAC with neural network approximation.

**Assumption 3** (Realizability)**.** *For $K \geq 1$, we assume that for all $\theta \in \Theta$, $Q_\lambda^{\pi_\theta} \in \mathcal{F}_{K,\bar{\nu},\mathcal{V}}$, where the function class $\mathcal{F}_{K,\bar{\nu},\mathcal{V}}$ is defined in Section 4.2.*

Note that $\mathcal{F}_{K,\bar{\nu},\mathcal{V}}$ approximates a rich class of functions over a compact space well for large $K$ (see Section 4.2). Also, Assumption 3 implies that there is a structure among the soft Q-functions in the policy class $\Theta$ since each $Q_\lambda^{\pi_\theta}$ can be written as a linear combination of $K$ functions that correspond to the transportation maps $v_k$. Thus, by the discussion in Section 4.2, if $Q_\lambda^{\pi_\theta}$ for the constrained parameter set $\theta \in \mathcal{B}_{m,R}^d(\theta(0))$ lies in a separable subspace of continuous functions, then the assumption holds for sufficiently large $K$ and $\bar{\nu}$ for some $\mathcal{V}$.

**Remark 3** (Realizability assumption for the actor)**.** *The main results in this work (i.e., Theorem 1) can be established without the realizability assumptions in Assumptions 2-3, which would bring function approximation errors $\epsilon_{\mathsf{app}}^{\mathsf{critic}}$ and $\epsilon_{\mathsf{app}}^{\mathsf{actor}}$, in both the critic and the actor, respectively. For the convergence results without the realizability assumptions, see Corollary 2.*

*The approximation error for the critic can be easily characterized by the infinite-width limit $\mathcal{F}_{\bar{\nu}}$, which was shown to be a universal approximator (Ji et al., 2019). Unlike the critic network, which learns only one function at each iteration of policy optimization, the actor network requires to approximate $Q^{\pi_t}$ for all $t = 0, 1, \ldots, T-1$ by a shared random initialization. This causes a significant challenge in characterizing the class of soft-Q functions that can be learned by using the actor network. To that end, the structure among $Q_\lambda^{\pi_t}$ is assumed in Assumption 3, which enables us to establish uniform approximation error bounds to handle the dynamic structure of the policy optimization over time steps. For further discussion on these uniform approximation results, see Section A.4 (particularly Remark 8).*

### 4.4.1 Performance Bounds under a Weak Distribution Mismatch Condition

First, we establish sample complexity and overparameterization bounds under a weak distribution mismatch condition, which is provided below. This condition is significantly weaker compared to the existing literature

(see, e.g., Wang et al. (2019); Liu et al. (2019); Agarwal et al. (2020)) as we proved that the policies achieve sufficient exploration by Proposition 2 (see Remark 4 for details).

**Assumption 4** (Weak distribution mismatch condition). *There exists a constant $C_\infty < \infty$ such that*

$$\sup_{t \geq 0} \ \mathbb{E}_{s \sim d_\mu^{\pi_t}} \left[ \left( \frac{d_\mu^{\pi^*}(s)}{d_\mu^{\pi_t}(s)} \right)^2 \right] \leq C_\infty^2.$$

**Remark 4** (Weak distribution mismatch condition). Note that a sufficient condition for Assumption 4 is an exploratory initial state distribution $\mu$, which covers the support of the state visitation distribution of $d_\mu^{\pi^*}$:

$$\sup_{s \in \mathsf{supp}(d_\mu^{\pi^*})} \frac{d_\mu^{\pi^*}(s)}{\mu(s)} < \infty, \tag{36}$$

since $\sqrt{\mathbb{E}_{s \sim d_\mu^{\pi_t}} \left[ \left( \frac{d_\mu^{\pi^*}(s)}{d_\mu^{\pi_t}(s)} \right)^2 \right]} \leq \frac{1}{1-\gamma} \left\| \frac{d_\mu^{\pi^*}}{\mu} \right\|_\infty$. Hence, if the initial distribution has a sufficiently large support set, then Assumption 4 is satisfied without any assumptions on $\{\pi_t : t \geq 0\}$. Together with Proposition 2, it *ensures* stability of the policy optimization with minimal assumptions on $\mu$, as Bhandari & Russo (2019) indicates that the condition $\|d_\mu^{\pi^*}/\mu\|_\infty < \infty$ is indeed necessary for convergence.

The following theorem is one of the main results in this paper, which establishes the convergence bounds of the NAC algorithm.

**Theorem 1** (Global Optimality and Convergence). Let $m' = \widetilde{O}\left( \frac{\bar{\nu}^4}{\varepsilon^2} \right)$, $T' = O\left( \frac{(1+2\bar{\nu})^2 \bar{\nu}^2}{\varepsilon^4} \right)$ as specified in Lemma 2. Under Assumptions 1-4, Algorithm 2 with $R > \bar{\nu}$ and regularization coefficient $\lambda > 0$ satisfies the following bounds:

(1) with step-size $\eta_t = \frac{1}{\lambda(t+1)}$, $t \geq 0$, we have

$$(1-\gamma) \min_{t \in [T]} \ \mathbb{E}[(V_\lambda^{\pi^*}(\mu) - V_\lambda^{\pi_t}(\mu))\mathbb{1}_A] \leq \frac{2R^2(1 + \log T)}{\lambda T} + 2R\sqrt{\rho_0} + 4\rho_0 T \lambda + M_\infty \left( \rho_1 + \varepsilon + \frac{Rq_{max}}{N^{1/4}} \right),$$

(2) with step-size $\eta_t = \eta \in (0, 1/\lambda)$, we have

$$(1-\gamma) \min_{t \in [T]} \ \mathbb{E}[(V_\lambda^{\pi^*}(\mu) - V_\lambda^{\pi_t}(\mu))\mathbb{1}_A] \leq \frac{\lambda e^{-\eta\lambda T} \log |\mathcal{A}|}{1 - e^{-\eta\lambda T}} + 2R\sqrt{\rho_0} + 4\rho_0 T \lambda + M_\infty \left( \rho_1 + \varepsilon + \frac{Rq_{max}}{N^{1/4}} \right) + 2\eta R^2,$$

for any $\delta \in (0, 1/3)$ where $\mathbb{P}(A) \geq 1 - 3\delta$ over the random initialization of the actor and critic networks, where $M_\infty = C_\infty(1 + \pi_{min}^{-1})$, $\rho_0 = \frac{16R}{\lambda\sqrt{m}} \left( \frac{R}{\lambda} + \sqrt{\log(1/\delta)} + \sqrt{d \log(m)} \right)$, $\rho_1 = \frac{16\bar{\nu}}{\sqrt{m}} \left( (d \log(m))^{\frac{1}{4}} + \sqrt{\log(K/\delta)} \right)$, and $q_{max} = 4(2R - \lambda \log \pi_{min})$.

Below we characterize the sample complexity, iteration complexity and overparameterization bounds based on Theorem 1.

**Corollary 1** (Sample Complexity and Overparameterization Bounds). For any $\epsilon > 0$ and $\delta \in (0, 1/3)$, Algorithm 2 with $R > \bar{\nu}$ satisfies:

$$\min_{t \in [T]} \ \mathbb{E}[(V_\lambda^{\pi^*}(\mu) - V_\lambda^{\pi_t}(\mu))\mathbb{1}_A] \leq \epsilon,$$

where $\mathbb{P}(A) \geq 1 - 3\delta$ over the random initialization of the actor-critic networks for the following parameters:

- *iteration complexity:* $T = \tilde{O}\left( \frac{R^2}{(1-\gamma)\lambda\epsilon} \right)$,

- *actor network width:* $m = \tilde{O}\left( \frac{R^8}{(1-\gamma)^4 \lambda^4 \epsilon^4} + \frac{R^6 \log(1/\delta)}{\lambda^2 (1-\gamma)^4 \epsilon^4} + \frac{M_\infty \bar{\nu}^2 \log(K/\delta)}{\epsilon^2 (1-\gamma)^2} \right)$,

- *critic sample complexity:* $T' = O\Big(\frac{M_\infty^2 R^4}{(1-\gamma)^2 \epsilon^4}\Big),$

- *critic network width:* $m' = \tilde{O}\Big(\frac{M_\infty^2 R^4 \log(1/\delta)}{(1-\gamma)^2 \epsilon^2}\Big),$

- *actor sample complexity:* $N = O\Big(\frac{M_\infty^4 R^4 q_{max}^4}{\epsilon^4 (1-\gamma)^4}\Big).$

Hence, the overall sample complexity of the Neural NAC algorithm is $\tilde{O}\Big(\frac{1}{\epsilon^5}\Big)$.

**Remark 5** (Bias-variance tradeoff in policy optimization). By Proposition 2, the network parameters evolve such that

$$\sup_{t \geq 0} \max_{i \in [m]} \|\theta_i(0) - \theta_i(t)\|_2 \leq \frac{R}{\lambda \sqrt{m}},$$

and $\sup_{t \geq 0} \sup_{s,a} \pi_t(a|s) < 1$. Hence, the NAC always performs a policy search within the class of randomized policies, which leads to fast and stable convergence under minimal regularity conditions. In particular, Assumption 4 is the mildest distributional mismatch condition in on-policy NPG/NAC settings to the best of our knowledge, and it suffices to establish convergence results in Theorem 1. On the other hand, entropy regularization introduces a bias term controlled by $\lambda$, hence the convergence is in the regularized MDP. Another way to see this is that deterministic policies, which require $\lim_t \|\theta(t)\|_2 = \infty$, may not be achieved for $\lambda > 0$ since $\theta(t)$ is always contained within a compact set. Letting $\lambda \downarrow 0$ eliminates the bias, but at the same time reduces the convergence speed and may lead to instability due to lack of exploration. Hence, there is a bias-variance tradeoff in policy optimization, controlled by $\lambda > 0$.

**Remark 6** (Different network widths for actor and critic). Corollary 1 indicates that the actor network requires $\tilde{O}(1/\epsilon^4)$ neurons while the critic network requires $\tilde{O}(1/\epsilon^2)$ although both approximate (soft) state-action value functions. This difference is because the actor network is required to uniformly approximate all state-action value functions over the trajectory, while the critic network approximates (pointwise) a single state-action value function at each iteration.

**Remark 7** (Fast initial convergence rate under constant step-sizes). The second part of Theorem 1 indicates that the convergence rate is $e^{-\Omega(T)}$ under a constant step-size $\eta \in (0, 1/\lambda)$, while there is an additional error term $2\eta R^2$. This justifies the common practice of "halving the step-size" in optimization (see, e.g., Karimi et al. (2016)) for the specific case of natural actor-critic that we investigate: one achieves a fast convergence rate with a constant step-size until the optimization stalls, then the process is repeated after halving the step-size.

The convergence bounds in Theorem 1 avoid the approximation errors that stem from the use of neural networks in the actor and the critic by Assumptions 2-3. In the following, we characterize the impact of these approximation errors on the performance of the algorithm. To that end, let

$$\epsilon_{\mathsf{app}}^{\mathsf{critic}} = \max_{0 \leq t < T} \mathbb{E} \min_{f \in \mathcal{F}_{\bar{\nu}}} \sqrt{\mathbb{E}_{s,a}[(f(s,a) - q_\lambda^{\pi_t}(s,a))^2]}, \tag{37}$$

$$\epsilon_{\mathsf{app}}^{\mathsf{actor}} = \max_{0 \leq t < T} \mathbb{E} \min_{u \in \mathcal{B}_{m,R}^d(0)} \sqrt{\mathbb{E}_{s,a}[(\nabla^\top f_0(s,a)u - Q_\lambda^{\pi_t}(s,a))^2]}, \tag{38}$$

be the approximation errors for the critic and the actor, respectively, where the outer expectation is over the parameter $\theta(t)$ given the random initialization. Note that $\nabla f_0(s,a)u$ is the linear approximation of $f(s,a; (c, u+\theta(0)))$ around the initialization (see Prop. 3). We have the following result.

**Corollary 2** (Global Optimality and Convergence). Let $m' = \tilde{O}\Big(\frac{\bar{\nu}^4}{\varepsilon^2}\Big)$, $T' = O\Big(\frac{(1+2\bar{\nu})^2 \bar{\nu}^2}{\varepsilon^4}\Big)$. Under Assumptions 1 and 4, Algorithm 2 with $R > \bar{\nu}$, $\lambda > 0$ and step-size $\eta_t = \frac{1}{\lambda(t+1)}$ satisfies

$$(1-\gamma) \min_{t \in [T]} \mathbb{E}[(V_\lambda^{\pi^*}(\mu) - V_\lambda^{\pi_t}(\mu))\mathbf{1}_A] \leq \frac{2R^2(1+\log T)}{\lambda T} + 2R\sqrt{\rho_0} + 4\rho_0 T\lambda$$

$$+ 4M_\infty \Big(\epsilon_{\mathsf{app}}^{\mathsf{actor}} + \epsilon_{\mathsf{app}}^{\mathsf{critic}} + \varepsilon + \frac{Rq_{max}}{N^{1/4}}\Big),$$

where $M_\infty = C_\infty(1 + \pi_{min}^{-1})$ and $\rho_0 = \frac{16R}{\lambda\sqrt{m}}\left(\frac{R}{\lambda} + \sqrt{\log(1/\delta)} + \sqrt{d\log(m)}\right)$ for any $\delta \in (0, 1/3)$ where $\mathbb{P}(A) \geq 1 - 3\delta$ over the random initialization of the actor and critic networks.

Corollary 2 explicitly shows the impact of the approximation errors for the actor and the critic.

### 4.4.2 Performance Bounds under a Strong Distribution Mismatch Condition

In the following, we consider the standard distribution mismatch condition (e.g., Liu et al. (2019); Wang et al. (2019)) and establish sample complexity and overparameterization bounds based on Theorem 1, for the unregularized MDP.

**Assumption 4'** (Strong distribution mismatch condition). *There exists a constant $\tilde{C}_\infty < \infty$ such that*

$$\sup_{t\geq 0} \mathbb{E}_{(s,a)\sim d_\mu^{\pi_t}\otimes\pi_t(\cdot|s)}\left[\left(\frac{d_\mu^{\pi^*}(s)\pi^*(a|s)}{d_\mu^{\pi_t}(s)\pi_t(a|s)}\right)^2\right] \leq \tilde{C}_\infty^2. \tag{39}$$

Note that Assumption 4' implies Assumption 4, and it is a considerably stronger assumption that necessitates our policies $\{\pi_t : t = 0, 1, \ldots, T - 1\}$ being sufficiently exploratory throughout policy optimization.

**Corollary 3.** Under Assumptions 1-3 and 4', for any $\epsilon > 0$ and $\delta \in (0, 1/3)$, Algorithm 2 with $R > \bar{\nu}$ and $\lambda = O(1/\sqrt{T})$ satisfies:

$$\min_{t\in[T]} \mathbb{E}[(\max_\pi V^\pi(\mu) - V^{\pi_t}(\mu))\mathbb{1}_A] \leq \epsilon,$$

where $\mathbb{P}(A) \geq 1 - 3\delta$ over the random initialization of the actor-critic networks for the following parameters:

- *iteration complexity:* $T = \tilde{O}\left(\frac{R^2}{(1-\gamma)\epsilon^2}\right)$,

- *actor network width:* $m = \tilde{O}\left(\frac{R^8}{(1-\gamma)^4\epsilon^8} + \frac{R^6\log(1/\delta)}{(1-\gamma)^4\epsilon^6} + \frac{\tilde{C}_\infty\bar{\nu}^2\log(K/\delta)}{\epsilon^2(1-\gamma)^2}\right)$,

- *critic sample complexity:* $T' = O\left(\frac{\tilde{M}_\infty^2 R^4}{(1-\gamma)^2\epsilon^4}\right)$,

- *critic network width:* $m' = \tilde{O}\left(\frac{\tilde{M}_\infty^2 R^4\log(1/\delta)}{(1-\gamma)^2\epsilon^2}\right)$,

- *actor sample complexity:* $N = O\left(\frac{\tilde{M}_\infty^4 R^4 q_{max}^4}{\epsilon^4(1-\gamma)^4}\right)$,

where $\tilde{M}_\infty = \tilde{C}_\infty(1 + \pi_{min}^{-1})$.

Hence, the overall sample complexity of Neural NAC for finding an $\epsilon$-optimal policy of the *unregularized* MDP is $\tilde{O}\left(\frac{1}{\epsilon^6}\right)$.

### 4.5 Comparison With Prior Work

Among the existing work that theoretically investigate policy gradient methods, the most related one is Wang et al. (2019), which considers Neural PG/NPG methods equipped with a two-layer neural network. We point key differences between our work and prior work:

- Prior work do not incorporate entropy regularization. As a result, they need a stronger concentrability coefficient assumption like Assumption 4' instead of the weaker Assumption 4 under which we are able to prove our main results.

- In the proofs in Appendix A, we will observe that one needs uniform function approximation bounds for the linearized actor network at finite-width in approximating soft Q-functions throughout policy optimization steps. We show that the problem structure, which will impose shared features among

| Algorithm | Width of actor, critic | Sample comp. | Error | Condition | Objective |
|---|---|---|---|---|---|
| Neural NPG (Wang et al., 2019) | $O(1/\epsilon^{12})$, $O(1/\epsilon^{12})$ | $O(1/\epsilon^{14})$ | $\epsilon + \epsilon_0$[1] | Strong | Unregularized |
| Neural NAC (ours) | $\tilde{O}(1/\epsilon^4)$, $\tilde{O}(1/\epsilon^2)$ | $\tilde{O}(1/\epsilon^5)$ | $\epsilon$ | Weak | Regularized |
| Neural NAC (ours) | $\tilde{O}(1/\epsilon^8)$, $\tilde{O}(1/\epsilon^2)$ | $\tilde{O}(1/\epsilon^6)$ | $\epsilon$ | Strong | Unregularized |

Table 1: The overparameterization and sample complexity bounds for variants of natural policy gradient with neural network approximation.

> soft Q-functions throughout policy optimization steps, will be of critical importance to achieve good performance. To address this challenge, which was not addressed in the prior work, we devise new techniques.

- While our algorithm is similar in spirit to the algorithms analyzed in the prior works, we also incorporate a number of important algorithmic ideas that are used in practice (e.g., entropy regularization, weight decay, gradient clipping). As a result, we have to use different analysis techniques. As a consequence of these algorithmic and analytical techniques, we obtain considerably sharper sample complexity and overparameterization bounds (see Table 1). Interestingly, all of these algorithmic improvements to the original NAC algorithms seem to be important to obtain the sharper bounds.

- We employ a symmetric initialization scheme proposed by Bai & Lee (2019) to ensure that $f_0(s, a) = 0$ for all $s, a$ despite the random initialization. As a consequence of symmetric initialization, we eliminate the impact of $f_0$ in the infinite width limit, which is effectively a noise term $\epsilon_0$ in the performance bounds (Wang et al., 2019; Liu et al., 2019; Cai et al., 2019).

## 5 Conclusion

In this paper, we established global convergence of the two-timescale entropy-regularized NAC algorithm with neural network approximation. We observed that entropy regularization, in combination with max-norm gradient clipping and weight-decay, led to significantly improved sample complexity and overparameterization bounds under weaker conditions since it (i) encourages exploration, (ii) controls the movement of the neural network parameters. We characterized the bias due to function approximation and sample-based estimation, and showed that overparameterization and increasing sample-size eliminates bias. Our analysis revealed the significant difference between the actor and critic in terms of the approximation and statistical errors, as the actor network to approximate all soft-Q functions throughout the policy optimization steps, requiring larger sample complexity and representation power. To that end, the extension of our work with a finer characterization of the uniform approximation error for the actor network, which would potentially lead to a larger hypothesis class than $\mathcal{F}_{K,\bar{\nu},\mathcal{V}}$, is an interesting direction for neural policy optimization.

In practice, single-timescale natural policy gradient methods are predominantly used in conjunction with entropy regularization and off-policy sampling (Haarnoja et al., 2018). The analysis techniques that we develop in this paper can be used to analyze these algorithms.

In supervised learning, softmax parameterization is predominantly used for multiclass classification problems, where natural gradient descent is employed for a better adjustment to the problem geometry (Goodfellow et al., 2016; Pascanu & Bengio, 2013; Zhang et al., 2019). The techniques that we developed in this paper can be useful in establishing convergence results and understanding the role of entropy regularization as well.

**Acknowledgments**

Niao He acknowledges support from Swiss National Science Foundation (SNSF) Project Funding No. 200021-207343 and SNSF Starting Grant. R. Srikant's research was supported in part by AFOSR Grant FA9550-24-1-0002, ONR Grant N00014-19-1-2566, and NSF Grants CNS 23-12714, CNS 21-06801, CCF 19-34986,

---

[1]$\epsilon_0$ is an error term that stems from the infinite-width limit of the neural network at random initialization, which is non-vanishing with $m$ or $T$.

and CCF 22-07547. S. Cayci would like to thank Siddhartha Satpathi for discussions on the proof of Lemma 7.

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

## A  Finite-Time Analysis of Neural NAC

In this section, we provide the convergence analysis of the algorithm.

### A.1  Analysis of Neural Network at Initialization

For $\delta \in (0,1)$ and any $R_0 > 0$, let

$$\rho_0(R_0, m, \delta) = \frac{16R_0}{\sqrt{m}}\Big(R_0 + \sqrt{\log(1/\delta)} + \sqrt{d\log(m)}\Big), \tag{40}$$

and define

$$A_0 = \Big\{ \sup_{x:\|x\|_2 \leq 1} \frac{R_0}{m} \sum_{i=1}^{m} \mathbb{1}\Big\{|\theta_i^\top(0)x| \leq \frac{R_0}{\sqrt{m}}\Big\} \leq \rho_0(R_0, m, \delta)\Big\}. \tag{41}$$

The following lemma bounds the deviation of the neural network from its linear approximation around the initialization, and it will be used throughout the convergence analysis.

**Lemma 3.** *Let $\theta_i(0) \sim \mathcal{N}(0, I_d)$ for all $i \in [m]$, $\theta \in \mathcal{B}^d_{m,R_0}(\theta(0))$ and $\theta' \in \mathcal{B}^d_{m,R_0}(0)$ for some $R_0 > 0$. Then,*

$$\sup_{x \in \mathbb{R}^d : \|x\|_2 \leq 1} \frac{1}{\sqrt{m}} \sum_{i=1}^{m} \left| \left( \mathbb{1}_{\{\theta_i^\top x \geq 0\}} - \mathbb{1}_{\{\theta_i^\top(0)x \geq 0\}} \right) \theta_i^\top(0)x \right| \leq \rho_0(R_0, m, \delta), \tag{42}$$

$$\sup_{x \in \mathbb{R}^d : \|x\|_2 \leq 1} \frac{1}{\sqrt{m}} \sum_{i=1}^{m} \left| \left( \mathbb{1}_{\{\theta_i^\top x \geq 0\}} - \mathbb{1}_{\{\theta_i^\top(0)x \geq 0\}} \right) \theta_i^\top x \right| \leq \rho_0(R_0, m, \delta), \tag{43}$$

$$\sup_{x \in \mathbb{R}^d : \|x\|_2 \leq 1} \frac{1}{\sqrt{m}} \sum_{i=1}^{m} \left| \left( \mathbb{1}_{\{\theta_i^\top x \geq 0\}} - \mathbb{1}_{\{\theta_i^\top(0)x \geq 0\}} \right) x^\top \theta_i' \right| \leq \rho_0(R_0, m, \delta), \tag{44}$$

*under the event $A_0$ defined in equation 41, which holds with probability at least $1 - \delta$ over the random initialization of the actor.*

*Proof.* Let $\Omega_d = \{x \in \mathbb{R}^d : \|x\|_2 \leq 1\}$. For $x \in \Omega_d$, let

$$S(x, \theta) = \left\{ i \in [m] : \mathbb{1}_{\{\theta_i^\top x \geq 0\}} \neq \mathbb{1}_{\{\theta_i^\top(0)x \geq 0\}} \right\}.$$

For any $i \in S(x, \theta)$, the following is true:

$$|\theta_i^\top(0)x| \leq |\theta_i^\top(0)x - \theta_i^\top x| \leq \|\theta_i - \theta_i(0)\|_2, \tag{45}$$

where the first inequality is true since $sign(\theta_i^\top(0)x) \neq sign(\theta_i^\top x)$ and the second inequality follows from Cauchy-Schwarz inequality and $x \in \Omega_d$. Therefore,

$$S(x, \theta) \subset \{i \in [m] : |\theta_i^\top(0)x| \leq \|\theta_i - \theta_i(0)\|_2\}.$$

Since

$$\frac{1}{\sqrt{m}} \sum_{i=1}^{m} \left| \left( \mathbb{1}_{\{\theta_i^\top x \geq 0\}} - \mathbb{1}_{\{\theta_i^\top(0)x \geq 0\}} \right) \theta_i^\top(0)x \right| = \frac{1}{\sqrt{m}} \sum_{i \in S(x, \theta)} |\theta_i^\top(0)x|,$$

we have:

$$\frac{1}{\sqrt{m}} \sum_{i=1}^{m} \left| \left( \mathbb{1}_{\{\theta_i^\top x \geq 0\}} - \mathbb{1}_{\{\theta_i^\top(0)x \geq 0\}} \right) \theta_i^\top(0)x \right| \leq \frac{1}{\sqrt{m}} \sum_{i=1}^{m} \mathbb{1}_{\{|\theta_i^\top(0)x| \leq \|\theta_i - \theta_i(0)\|_2\}} \|\theta_i - \theta_i(0)\|_2.$$

Since $\max_{i \in [m]} \|\theta_i - \theta_i(0)\|_2 \leq \frac{R_0}{\sqrt{m}}$, the above inequality leads to the following:

$$\frac{1}{\sqrt{m}} \sum_{i=1}^{m} \left| \left( \mathbb{1}_{\{\theta_i^\top x \geq 0\}} - \mathbb{1}_{\{\theta_i^\top(0)x \geq 0\}} \right) \theta_i^\top(0)x \right| \leq \frac{R_0}{m} \sum_{i=1}^{m} \mathbb{1}_{\{|\theta_i^\top(0)x| \leq R_0/\sqrt{m}\}}.$$

Taking supremum over $x \in \Omega_d$, and using Lemma 4 in Satpathi et al. (2020) on the RHS of the above inequality concludes the proof.

In order to prove equation 43, similar to equation 45, we have the following inequality:

$$|\theta_i^\top x| \leq |\theta_i^\top x - \theta_i^\top(0)x| \leq \|\theta_i - \theta_i(0)\|_2.$$

Using this, the proof follows from exactly the same steps. □

Note that Lemma 3 is an extension of the concentration bounds in Ji & Telgarsky (2019); Du et al. (2018); Satpathi et al. (2020) for neural networks. On the other hand, our concentration result provides uniform convergence over $\Omega_d = \{x \in \mathbb{R}^d : \|x\|_2 \leq 1\}$ rather than finitely many points, thus it is a stronger concentration bound compared to the ones in the literature, which are used to analyze neural networks (Ji & Telgarsky, 2019; Du et al., 2018). We need these uniform concentration inequalities to address the challenges due to the dynamics policy optimization, e.g., distributional shift.

## A.2 Impact of Entropy Regularization

First, we analyze the impact of entropy regularization, which will yield key results in the convergence analysis.

*Proof of Proposition 2.* Recall from Line 2 in Algorithm 3 that the policy update is as follows:

$$\theta(t+1) = \theta(t) + \eta_t u_t - \eta_t \lambda(\theta(t) - \theta(0)).$$

Let $\bar{\theta}(t) = \theta(t) - \theta(0)$ for all $t \geq 0$. Then, the update rule can be written as:

$$\bar{\theta}(t+1) = \bar{\theta}(t)(1 - \eta_t \lambda) + \eta_t u_t.$$

Since the step-size is $\eta_t \lambda = \frac{1}{t+1}$, we have:

$$\bar{\theta}(t+1) = \frac{1}{\lambda(t+1)} \sum_{k=0}^{t} u_k,$$

by induction. Hence, by triangle inequality:

$$\|\bar{\theta}_i(t+1)\|_2 = \|\theta_i(t+1) - \theta_i(0)\|_2 \leq \frac{1}{\lambda(t+1)} \sum_{k=0}^{t} \|u_{i,k}\|_2, \tag{46}$$

for any $i \in [m]$. Note that $u_k \in \mathcal{B}_{m,R}^d(0)$ as a consequence of projection, therefore $\|u_{i,k}\| \leq R/\sqrt{m}$ for all $i \in [m]$. Hence, by equation 46, we conclude that

$$\max_{i \in [m]} \|\theta_i(t) - \theta_i(0)\|_2 \leq \frac{R}{\lambda\sqrt{m}}, \tag{47}$$

for any $t \geq 0$. Also, since $w_t = u_t - \lambda(\theta(t) - \theta(0))$, we have:

$$\sup_{t \geq 0} \|w_t\|_2 \leq \|u_t\|_2 + \lambda\|\theta(t) - \theta(0)\|_2 \leq 2R. \tag{48}$$

Under a constant step-size $\eta \in (0, 1/\lambda)$, we can expand the parameter movement for any $t \geq 1$ as follows:

$$\bar{\theta}_i(t+1) = \bar{\theta}_i(t) \cdot (1 - \eta\lambda) + \eta \cdot u_{i,t},$$

$$= \bar{\theta}_i(t-1) \cdot (1 - \eta\lambda)^2 + \eta(1 - \lambda\eta)u_{i,t-1} + \eta u_{i,t}, \qquad \vdots$$

$$= \bar{\theta}_i(0)(1 - \eta\lambda)^t + \eta \sum_{k=0}^{t}(1 - \eta\lambda)^k u_{i,t-k} = \eta \sum_{k=0}^{t}(1 - \eta\lambda)^k u_{i,t-k},$$

for any neuron $i \in [m]$. Then, we have:

$$\|\theta_i(t+1) - \theta_i(0)\|_2 \leq \eta \sum_{k=0}^{t}(1 - \eta\lambda)^k \|u_{i,k}\|_2 \leq \frac{R}{\lambda\sqrt{m}}(1 - (1 - \eta\lambda)^{t+1}),$$

$$\leq \frac{R}{\lambda\sqrt{m}}, \tag{49}$$

which follows from triangle inequality, $\|u_{i,k}\|_2 \leq R/\sqrt{m}$ due to the projection, and the fact that $(1 - (1 - \eta\lambda)^t) \leq 1$ for any $t \geq 0$.

In order to prove the lower bound for $\inf_{t \geq 0, (s,a) \in \mathcal{S} \times \mathcal{A}} \pi_t(a|s)$, first recall that $\pi_t(a|s) \propto \exp(f_t(s,a))$. Hence, a uniform upper bound on $|f_t(s,a)|$ over all $t \geq 0$ and $(s,a) \in \mathcal{S} \times \mathcal{A}$ suffices to lower bound $\pi_t(a|s)$. By symmetric initialization, $f_0(s,a) = 0$ for all $(s,a) \in \mathcal{S} \times \mathcal{A}$. Hence,

$$f_t(s,a) = \frac{1}{\sqrt{m}} \sum_{i=1}^{m} c_i \left([\theta_i(t) - \theta_i(0)]^\top (s,a) \mathbb{1}_{\{\theta_i^\top(t)(s,a) \geq 0\}}\right)$$

$$+ \frac{1}{\sqrt{m}} \sum_{i=1}^{m} c_i (\mathbb{1}_{\{\theta_i^\top(s,a) \geq 0\}} - \mathbb{1}_{\{\theta_i^\top(0)(s,a) \geq 0\}})\theta_i^\top(t)(s,a). \tag{50}$$

First, we bound the first summand on the RHS of equation 50 by using equation 47 and triangle inequality:

$$\sup_{s,a} \left| \frac{1}{\sqrt{m}} \sum_{i=1}^{m} c_i \left( [\theta_i(t) - \theta_i(0)]^\top (s,a) \mathbb{1}_{\{\theta_i^\top (t)(s,a) \geq 0\}} \right) \right| \leq \frac{R}{\lambda}, \tag{51}$$

since $|c_i \mathbb{1}_{\{\theta_i^\top (t)(s,a) \geq 0\}} \cdot (s,a)| \leq 1$. For the last term in equation 50, first note that $\max_{i \in [m]} \|\theta_i(t) - \theta_i(0)\|_2 \leq \frac{R}{\lambda \sqrt{m}}$, so we can use Lemma 3. By using triangle inequality and Lemma 3:

$$\frac{1}{\sqrt{m}} \sum_{i=1}^{m} \left| (\mathbb{1}_{\{\theta_i^\top (s,a) \geq 0\}} - \mathbb{1}_{\{\theta_i^\top (0)(s,a) \geq 0\}}) \theta_i^\top (t)(s,a) \right| \leq \rho_0 \left( \frac{R}{\lambda}, m, \delta \right),$$

with probability at least $1 - \delta$ over the random initialization of the actor network. Hence, with probability at least $1 - \delta$,

$$\sup_{s,a} |f_t(s,a)| \leq R/\lambda + \rho_0 \left( \frac{R}{\lambda}, m, \delta \right),$$

and $\pi_t(a|s) \geq \frac{1}{|\mathcal{A}|} e^{\frac{-2R}{\lambda} - 2\rho_0 \left( \frac{R}{\lambda}, m, \delta \right)}$.

$\square$

## A.3   Lyapunov Drift Analysis

First, we present a key lemma which will be used throughout the analysis.

**Lemma 4** (Log-linear approximation error). *Let*

$$\widetilde{\pi}_t(a|s) = \frac{\exp(\nabla_\theta^\top f_0(s,a)\theta(t))}{\sum_{a' \in \mathcal{A}} \exp(\nabla_\theta^\top f_0(s,a')\theta(t))},$$

*be log-linear approximation of the policy $\pi_t(a|s)$. Then, for any $\delta \in (0,1)$, we have:*

$$\sup_{t \geq 0} \sup_{s,a} \left| \log \frac{\widetilde{\pi}_t(a|s)}{\pi_t(a|s)} \right| \leq 3\rho_0 \left( \frac{R}{\lambda}, m, \delta \right), \tag{52}$$

*over $A_0$.*

*Proof.* Note that $f_t(s,a) = \nabla f_t(s,a)\theta(t)$ for a ReLU neural network. By using this, we can write the log-linear approximation error as follows:

$$\left| \log \frac{\widetilde{\pi}_t(a|s)}{\pi_t(a|s)} \right| \leq |(\nabla f_t(s,a) - \nabla f_0(s,a))^\top \theta(t)| + \left| \log \frac{\sum_{a'} e^{\nabla^\top f_0(s,a')\theta(t)} e^{(\nabla f_t(s,a') - \nabla f_0(s,a'))^\top \theta(t)}}{\sum_{a'} e^{\nabla^\top f_0(s,a')\theta(t)}} \right|. \tag{53}$$

By log-sum inequality (Theorem 2.7.1 in Cover & Thomas (2006)), for any $x_a, y_a > 0$,

$$\log \frac{\sum_a x_a}{\sum_a y_a} \leq \sum_a \frac{x_a}{\sum_{a'} x_{a'}} \log \frac{x_a}{y_a}.$$

Setting $x_a = e^{\nabla^\top f_0(s,a)\theta(t)}$ and $y_a = e^{\nabla^\top f_0(s,a)\theta(t)} e^{(\nabla f_t(s,a) - \nabla f_0(s,a))^\top \theta(t)}$, we have:

$$\log \frac{\sum_{a'} e^{\nabla^\top f_0(s,a')\theta(t)}}{\sum_{a'} e^{\nabla^\top f_t(s,a')\theta(t)}} \leq \sum_{a'} \widetilde{\pi}_t(a'|s) |(\nabla f_t(s,a') - \nabla f_0(s,a'))^\top \theta(t)|. \tag{54}$$

Setting $y_a = e^{\nabla^\top f_0(s,a)\theta(t)}$ and $x_a = e^{\nabla^\top f_0(s,a)\theta(t)} e^{(\nabla f_t(s,a) - \nabla f_0(s,a))^\top \theta(t)}$, we have:

$$\log \frac{\sum_{a'} e^{\nabla^\top f_t(s,a')\theta(t)}}{\sum_{a'} e^{\nabla^\top f_0(s,a')\theta(t)}} \leq \sum_{a'} \pi_t(a'|s) |(\nabla f_t(s,a') - \nabla f_0(s,a'))^\top \theta(t)|. \tag{55}$$

Using equation 54 and equation 55 to bound the last term in equation 53, we obtain:

$$\left|\log\frac{\widetilde{\pi}_t(a|s)}{\pi_t(a|s)}\right| \le |(\nabla f_t(s,a) - \nabla f_0(s,a))^\top \theta(t)| + \sum_{a'}\left[\pi_t(a'|s) + \widetilde{\pi}_t(a'|s)\right]|(\nabla f_t(s,a') - \nabla f_0(s,a'))^\top \theta(t)|. \quad (56)$$

By Lemma 3, under the event $A_0$, $|(\nabla f_t(s,a) - \nabla f_0(s,a))^\top \theta(t)| \le \rho_0(R/\lambda, m, \delta)$ for all $t \ge 0, s \in \mathcal{S}, a \in \mathcal{A}$. Hence, under the event $A_0$, we have:

$$\sup_{t \ge 0}\ \sup_{s,a}\ \left|\log\frac{\widetilde{\pi}_t(a|s)}{\pi_t(a|s)}\right| \le 3\rho_0(R/\lambda, m, \delta),$$

which concludes the proof. $\qquad\square$

The following result is standard in the analysis of policy gradient methods (Kakade & Langford, 2002; Cayci et al., 2021).

**Lemma 5** (Lemma 5, (Cayci et al., 2021)). *For any $\theta, \theta' \in \mathbb{R}^d$ and $\mu$, we have:*

$$V_\lambda^{\pi_\theta}(\mu) - V_\lambda^{\pi_{\theta'}}(\mu) = \frac{1}{1-\gamma}\mathbb{E}_{s\sim d_\mu^{\pi_\theta}, a\sim\pi_\theta(\cdot|s)}\left[A_\lambda^{\pi_{\theta'}}(s,a) + \lambda\log\frac{\pi_{\theta'}(a|s)}{\pi_\theta(a|s)}\right], \quad (57)$$

*where $A_\lambda^{\pi_\theta}$ is the advantage function defined in equation 8.*

Lemma 5 is an extension of the performance difference lemma in Kakade & Langford (2002), and the proof can be found in Cayci et al. (2021). In the following, we provide the main Lyapunov drift, which is central to the proof. This Lyapunov function is widely used in the analysis of natural gradient descent algorithms (Peters & Schaal, 2008; Agarwal et al., 2020; Wang et al., 2019; Cayci et al., 2021).

**Definition 1** (Potential function). *For any policy $\pi \in \Pi$, the potential function $\Psi$ is defined as follows:*

$$\Psi(\pi) = \mathbb{E}_{s\sim d_\mu^{\pi^*}}\left[D_{KL}\left(\pi^*(\cdot|s)\|\pi(\cdot|s)\right)\right]. \quad (58)$$

**Lemma 6** (Lyapunov drift). *For any $t \ge 0$, let $\Delta_t = V_\lambda^{\pi^*}(\mu) - V_\lambda^{\pi_t}(\mu)$. Then,*

$$\begin{aligned}
\Psi(\pi_{t+1}) - \Psi(\pi_t) \le &-\eta_t\lambda\Psi(\pi_t) - \eta_t(1-\gamma)\Delta_t + 2\eta_t^2 R^2 \\
&+ \eta_t\mathbb{E}_{s\sim d_\mu^{\pi^*}, a\sim\pi_t(\cdot|s)}\left[\nabla^\top f_0(s,a)u_t - Q_\lambda^{\pi_t}(s,a)\right] \\
&- \eta_t\mathbb{E}_{s\sim d_\mu^{\pi^*}, a\sim\pi^*(\cdot|s)}\left[\nabla^\top f_0(s,a)u_t - Q_\lambda^{\pi_t}(s,a)\right] \\
&+ (\eta_t\lambda + 6)\rho_0(R/\lambda, m, \delta) + 2\eta_t R\sqrt{\rho_0(R/\lambda, m, \delta)},
\end{aligned} \quad (59)$$

*in the event $A_0$ which holds with probability at least $1 - \delta$ over the random initialization of the actor.*

*Proof.* First, note that the log-linear approximation of $\pi_\theta$ is smooth (Agarwal et al., 2020):

$$\|\nabla\log\widetilde{\pi}_\theta(a|s) - \nabla\log\widetilde{\pi}_{\theta'}(a|s)\|_2 \le \|\theta - \theta'\|_2, \quad (60)$$

for any $s, a$ since $\|\nabla f_0(s,a)\|_2 \le 1$. Also,

$$\Psi(\pi_{t+1}) - \Psi(\pi_t) = \mathbb{E}_{s\sim d_\mu^{\pi^*}, a\sim\pi^*(\cdot|s)}\left[\log\frac{\pi_t(a|s)}{\pi_{t+1}(a|s)}\right].$$

To use the smoothness of log-linear approximation, we use a telescoping sum and obtain:

$$\Psi(\pi_{t+1}) - \Psi(\pi_t) = \mathbb{E}_{s\sim d_\mu^{\pi^*}, a\sim\pi^*(\cdot|s)}\left[\log\frac{\widetilde{\pi}_t(a|s)}{\widetilde{\pi}_{t+1}(a|s)} + \log\frac{\pi_t(a|s)}{\widetilde{\pi}_t(a|s)} + \log\frac{\widetilde{\pi}_{t+1}(a|s)}{\pi_{t+1}(a|s)}\right].$$

By Lemma 4, the last two terms are bounded by $\rho_0(R/\lambda, m, \delta)$. Let

$$D_t = \mathbb{E}_{s \sim d_\mu^{\pi^*}, a \sim \pi^*(\cdot|s)} \left[ \log \frac{\widetilde{\pi}_t(a|s)}{\widetilde{\pi}_{t+1}(a|s)} \right].$$

Then, by the smoothness of the log-linear approximation, we have:

$$D_t \leq -\eta_t \mathbb{E}_{s \sim d_\mu^{\pi^*}, a \sim \pi^*(\cdot|s)} \nabla_\theta^\top \log \widetilde{\pi}_t(a|s) w_t + \frac{\eta_t^2 \|w_t\|_2^2}{2},$$

Recall $\Delta_t = V_\lambda^{\pi^*}(\mu) - V_\lambda^{\pi_t}(\mu)$. Using Lemma 5 and the definition of the advantage function, we obtain:

$$D_t \leq -\eta_t \lambda \Psi(\pi_t) - \eta_t(1-\gamma)\Delta_t - \eta_t \mathbb{E}_{s \sim d_\mu^{\pi^*}(s), a \sim \pi_t(a|s)} [Q_\lambda^{\pi_t}(s,a) - \lambda \log \pi_t(a|s)]$$

$$- \eta_t \mathbb{E}_{s \sim d_\mu^{\pi^*}, a \sim \pi^*(\cdot|s)} [\nabla^\top \log \widetilde{\pi}_t(a|s) w_t - q_\lambda^{\pi_t}(s,a)] + \frac{\eta_t^2 \|w_t\|_2^2}{2}. \tag{61}$$

Since we have $\nabla \log \widetilde{\pi}_t(a|s) = \nabla f_0(s,a) - \mathbb{E}_{a' \sim \widetilde{\pi}_t(\cdot|s)}[\nabla f_0(s,a')]$, we have the following inequality:

$$\begin{aligned}
D_t \leq &-\eta_t \lambda \Psi(\pi_t) - \eta_t(1-\gamma)\Delta_t \\
&+ \eta_t \mathbb{E}_{s \sim d_\mu^{\pi^*}(s), a \sim \pi_t(a|s)} [\nabla^\top f_0(s,a) w_t - Q_\lambda^{\pi_t}(s,a) + \lambda f_t(s,a)] \\
&- \eta_t \mathbb{E}_{s \sim d_\mu^{\pi^*}, a \sim \pi^*(\cdot|s)} [\nabla^\top f_0(s,a) w_t - Q_\lambda^{\pi_t}(s,a) + \lambda f_t(s,a)] \\
&+ \eta_t \mathbb{E}_{s \sim d_\mu^{\pi^*}} \left[ \sum_{a \in \mathcal{A}} (\widetilde{\pi}_t(a|s) - \pi_t(a|s)) \nabla^\top f_0(s,a) w_t \right] + \frac{\eta_t^2 \|w_t\|_2^2}{2}.
\end{aligned}$$

By the definition of $w_t = u_t - \lambda[\theta(t) - \theta(0)]$ and the fact that $f_0(s,a) = 0$ due to the symmetric initialization, we have:

$$\nabla^\top f_0(s,a) w_t = \nabla^\top f_0(s,a) u_t - \lambda \nabla^\top f_0(s,a) \theta(t).$$

Substituting this identity to the above inequality, we have:

$$\begin{aligned}
D_t \leq &-\eta_t \lambda \Psi(\pi_t) - \eta_t(1-\gamma)\Delta_t \\
&+ \eta_t \mathbb{E}_{s \sim d_\mu^{\pi^*}(s), a \sim \pi_t(a|s)} [\nabla^\top f_0(s,a) u_t - Q_\lambda^{\pi_t}(s,a)] \\
&- \eta_t \mathbb{E}_{s \sim d_\mu^{\pi^*}, a \sim \pi^*(\cdot|s)} [\nabla^\top f_0(s,a) u_t - Q_\lambda^{\pi_t}(s,a)] \\
&+ 2\eta_t R \mathbb{E}_{s \sim d_\mu^{\pi^*}} \left[ \sum_a |\widetilde{\pi}_t(a|s) - \pi_t(a|s)| \right] \\
&+ \eta_t \lambda \mathbb{E}_{s \sim d_\mu^{\pi^*}} \left[ \sum_{a \in \mathcal{A}} |\pi_t(a|s) - \pi^*(a|s)| \cdot (\nabla f_t(s,a) - \nabla f_0(s,a))^\top \theta(t) \right] + 2\eta_t^2 R^2,
\end{aligned} \tag{62}$$

where we used equation 48 to bound $\|w_t\|_2$. Furthermore, note that

$$|(\nabla f_t(s,a) - \nabla f_0(s,a))^\top \theta(t)| \leq \frac{1}{\sqrt{m}} \sum_{i=1}^m \left| \left( \mathbb{1}_{\{\theta_i^\top x \geq 0\}} - \mathbb{1}_{\{\theta_i^\top(0)x \geq 0\}} \right) \theta_i^\top(t) x \right|,$$

for any $x = (s,a)^\top \in \mathbb{R}^d$. Thus, by Lemma 3,

$$\sup_{s,a} |(\nabla f_t(s,a) - \nabla f_0(s,a))^\top \theta(t)| \leq \rho_0(R/\lambda, m, \delta).$$

This bounds the penultimate term in equation 62. Finally, in order to bound the fifth term in equation 62, we use Pinsker's inequality and then Lemma 4:

$$\sup_{s \in \mathcal{S}} \|\widetilde{\pi}_t(\cdot|s) - \pi_t(\cdot|s)\|_1 \leq \sup_{s \in \mathcal{S}} \sqrt{\sum_a \pi_t(a|s) \log \frac{\pi_t(a|s)}{\widetilde{\pi}_t(a|s)}} \leq \sqrt{\rho_0(R/\lambda, m, \delta)}.$$

Substituting these into equation 62 and then into equation 59, the desired result follows. $\qquad \square$

### A.4 Analysis of the Function Approximation Error: How Do Neural Networks Address Distributional Shift in Policy Optimization?

As a specific feature of reinforcement learning, policy optimization in particular, the probability distribution of the underlying system changes over time as a function of the control policy. Consequently, the function approximator (i.e., the actor network in our case) needs to adapt to this distributional shift throughout the policy optimization steps. In this subsection, we analyze the function approximation error, which sheds light on how neural networks in the NTK regime address the distributional shift challenge.

Now we focus on the error term in Lemma 6:

$$\epsilon_{bias}^{\pi_t} = \mathbb{E}_{s \sim d_\mu^*} \Big[ \sum_{a \in \mathcal{A}} \big( \pi_t(a|s) - \pi^*(a|s) \big) \big( \nabla^\top f_0(s,a) u_t - Q_\lambda^{\pi_t}(s,a) \big) \Big]. \tag{63}$$

Note that $\epsilon_{bias}^{\pi_t}$ can be equally expressed as follows:

$$\epsilon_{bias}^{\pi_t} = \mathbb{E}_{s \sim d_\mu^*} \Big[ \sum_{a \in \mathcal{A}} \big( \pi_t(a|s) - \pi^*(a|s) \big) \big( \nabla^\top \log \pi_t(a|s) u_t - \Xi_\lambda^{\pi_t}(s,a) \big) \Big]$$
$$+ \mathbb{E}_{s \sim d_\mu^{\pi^*}} \Big[ \sum_{a \in \mathcal{A}} \big( \pi_t(a|s) - \pi^*(a|s) \big) \Big( \big[ \nabla f_0(s,a) - \nabla f_t(s,a) \big]^\top u_t \Big) \Big],$$

where $\Xi_\lambda^\pi$ is the soft advantage function. The above identity provides intuition about the choice of sample-based gradient update $u_t$ in Algorithm 2, which we will investigate in detail later.

Let

$$L_0(u, \theta) = \mathbb{E}[(\nabla^\top f_0(s,a) u - Q_\lambda^{\pi_\theta}(s,a))^2].$$

In the following, we answer the following question: *given the perfect knowledge of the soft Q-function $Q_\lambda^{\pi_\theta}$, what is $\min_u L_0(u, \theta(t))$?*

**Proposition 3** (Approximation Error). *Under symmetric initialization of the actor network, we have the following results:*

- **Pointwise approximation error:** *For any $\theta \in \Theta$ and $Q_\lambda^{\pi_\theta} \in \mathcal{F}_{\bar{\nu}}$,*

$$\mathbb{E}\Big[ \min_{u \in \mathbb{R}^{m \times d}} L_0(u, \theta) \Big] \leq \frac{4\bar{\nu}^2}{m}, \tag{64}$$

*where the expectation is over the random initialization of the actor network.*

- **Uniform approximation error:** *Let*

$$A_1 = \Big\{ \sup_{\substack{s,a \\ \theta \in \Theta}} \min_u |\nabla^\top f_0(s,a) u - Q_\lambda^{\pi_\theta}(s,a)| \leq \frac{2\bar{\nu}}{\sqrt{m}} \Big( (d \log(m))^{\frac{1}{4}} + \sqrt{\log\Big(\frac{K}{\delta}\Big)} \Big) \Big\}.$$

*Then, under Assumption 3, $A_1$ holds with probability at least $1 - \delta$ over the random initialization of the actor network. Furthermore,*

$$\mathbb{E}\Big[ \mathbb{1}_{A_0 \cap A_1} \sup_\theta \min_u L_0(u, \theta) \Big] \leq \frac{4\bar{\nu}^2}{m} \Big( (d \log(m))^{\frac{1}{4}} + \sqrt{\log\Big(\frac{K}{\delta}\Big)} \Big)^2. \tag{65}$$

*Proof.* For a given policy parameter $\theta \in \Theta$, let the transportation mapping of $Q_\lambda^{\pi_\theta}$ be $v_\theta$ and let

$$Y_i^\theta(s,a) = v_\theta^\top(\theta_i(0))(s,a) \cdot \mathbb{1}_{\{\theta_i^\top(0)(s,a) \geq 0\}}, \ i \in [m].$$

Note that $\mathbb{E}[Y_i^\theta(s,a)] = Q_\lambda^{\pi_\theta}(s,a)$ for any $(s,a) \in \mathcal{S} \times \mathcal{A}$. Also, let

$$u_\theta^* = \Big[ \frac{1}{\sqrt{m}} c_i v_\theta(\theta_i(0)) \Big]_{i \in [m]}.$$

Since $u_\theta^* \in \mathcal{B}_{m,\bar\nu}^d(0)$ for all $\theta \in \Theta$, projected risk minimization within $\mathcal{B}_{m,R}^d(0)$ for $R \geq \bar\nu$ suffices for optimality. We have

$$\nabla^\top f_0(s,a) u_\theta^* = \frac{1}{m} \sum_{i=1}^m Y_i^\theta(s,a),$$

and

$$\min_u L_0(u,\theta) \leq \mathbb{E}_{s \sim d_\mu^{\pi_\theta}, a \sim \pi_\theta(\cdot|s)}\Big[\Big(\frac{1}{m}\sum_{i=1}^m Y_i^\theta(s,a) - \mathbb{E}[Y_1^\theta(s,a)]\Big)^2\Big]. \tag{66}$$

**1. Pointwise approximation error:** First we consider a given fixed $\theta \in \Theta$. Taking the expectation in equation 66 and using Fubini's theorem,

$$\mathbb{E}[\min_u L_0(u,\theta)] \leq \mathbb{E}_{s,a}\mathbb{E}\Big[\Big(\frac{1}{m}\sum_{i=1}^m Y_i^\theta(s,a) - \mathbb{E}[Y_1^\theta(s,a)]\Big)^2\Big],$$

$$= 2\mathbb{E}_{s,a}\Big[\frac{4}{m^2}\sum_{i=1}^{m/2} Var(Y_i^\theta(s,a)) + \frac{4}{m^2}\sum_{\substack{i,j=1\\i\neq j}}^{m/2} Cov(Y_i^\theta(s,a), Y_j^\theta(s,a))\Big], \tag{67}$$

$$= 4\mathbb{E}_{s,a}\Big[\frac{Var(Y_1^\theta(s,a))}{m^2}\Big], \tag{68}$$

$$\leq \frac{4}{m^2}\mathbb{E}_{s,a}\mathbb{E}[(Y_1^\theta(s,a))^2], \tag{69}$$

where the identity equation 67 is due to the symmetric initialization, equation 68 holds because $\{Y_i^\theta(s,a) : i = 1, 2, \ldots, m/2\}$ is independent (since $\{\theta_i(0) : i \in [m/2]\}$ is independent). By Cauchy-Schwarz inequality and the fact that $v_\theta \in \mathcal{H}_{\bar\nu}$, we have:

$$|Y_i^\theta(s,a)| \leq \|v_\theta(\theta_i(0))\|_2 \leq \bar\nu.$$

Hence, using this in equation 69, we obtain:

$$\mathbb{E}\min_u L_0(u,\theta) \leq \frac{4\bar\nu^2}{m^2}. \tag{70}$$

**2. Uniform approximation error:** For any $\theta \in \Theta$, since $Q_\lambda^{\pi_\theta} \in \mathcal{F}_{K,\bar\nu,\mathcal{V}}$ there exists $\alpha^\theta = (\alpha_1^\theta, \alpha_2^\theta, \ldots, \alpha_K^\theta) \in \mathbb{R}^K$ such that $\|\alpha^\theta\|_1 \leq 1$ and $v_\theta = \sum_k \alpha_k^\theta v_k$. We consider the following error:

$$R_m(\Theta) = \sup_{(s,a)\in\mathcal{S}\times\mathcal{A}} \sup_{\theta\in\Theta} \Big|\frac{1}{m}\sum_{i=1}^m Y_i^\theta(s,a) - \mathbb{E}[Y_1^\theta(s,a)]\Big|. \tag{71}$$

Then, we have the following identity from the definition of $v_\theta$:

$$R_m(\Theta) = \sup_{s,a} \sup_\theta \Big|\sum_{k=1}^K \alpha_k^\theta \cdot \Big(\frac{1}{m}\sum_{i=1}^m Z_i^k(s,a) - \mathbb{E}[Z_1^k(s,a)]\Big)\Big|, \tag{72}$$

where $Z_i^k(s,a) = v_k^\top(\theta_i(0))(s,a)\mathbb{1}\{\theta_i^\top(0)(s,a) \geq 0\}$. Then, by triangle inequality,

$$R_m(\Theta) \leq \sup_{s,a} \sup_\theta \max_{k\in[K]} \Big|\frac{1}{m}\sum_{i=1}^m Z_i^k(s,a) - \mathbb{E}[Z_1^k(s,a)]\Big| \cdot \|\alpha^\theta\|_1,$$

$$\leq \max_{k\in[K]} \sup_{s,a} \Big|\frac{1}{m}\sum_{i=1}^m Z_i^k(s,a) - \mathbb{E}[Z_1^k(s,a)]\Big|. \tag{73}$$

By using union bound and equation 73, for any $z > 0$, we have the following:

$$\mathbb{P}(R_m(\Theta) > z) \leq \sum_{k=1}^K \mathbb{P}\Big(\sup_{s,a} \Big|\frac{1}{m}\sum_{i=1}^m Z_i^k(s,a) - \mathbb{E}[Z_1^k(s,a)]\Big| > z\Big). \tag{74}$$

We utilize the following to obtain a uniform bound for $|\frac{1}{m}\sum_{i=1}^m Z_i^k(s,a) - \mathbb{E}[Z_1^k(s,a)]|$ over all $(s,a) \in \mathcal{S}\times\mathcal{A}$.

**Lemma 7.** *For any $k \in [K]$, for any $\delta \in (0,1)$, the following holds:*

$$\sup_{s,a} \left| \frac{1}{m} \sum_{i=1}^{m} Z_i^k(s,a) - \mathbb{E}[Z_1^k(s,a)] \right| \leq \frac{4\bar{\nu}(d\log m)^{1/4}}{\sqrt{m}} + \frac{4\bar{\nu}\sqrt{\log(1/\delta)}}{\sqrt{m}}, \tag{75}$$

*with probability at least $1 - \delta$.*

Hence, using Lemma 7 and equation 74 with $z = \frac{4\bar{\nu}(d\log m)^{1/4}}{\sqrt{m}} + \frac{4\bar{\nu}\sqrt{\log(K/\delta)}}{\sqrt{m}}$, we conclude that

$$R_m(\Theta) \leq \frac{4\bar{\nu}(d\log m)^{1/4}}{\sqrt{m}} + \frac{4\bar{\nu}\sqrt{\log(K/\delta)}}{\sqrt{m}},$$

with probability at least $1 - \delta$. The expectation result follows from this inequality. $\qquad\square$

Now, we have the following result for the approximation error under $\pi_t$.

**Corollary 4.** *Under Assumption 3, we have:*

$$\mathbb{E}[\mathbb{1}_{A_0 \cap A_1} \min_u L_0(u, \theta(t))] \leq \frac{16\bar{\nu}^2}{m}\left((d\log(m))^{\frac{1}{4}} + \sqrt{\log\left(\frac{K}{\delta}\right)}\right)^2,$$

*where the event $A_1$, defined in Proposition 3, holds with probability at least $1-\delta$ over the random initialization of the actor.*

**Remark 8 (Why do we need a uniform approximation error bound?).** Note that for a given fixed policy $\pi_\theta, \theta \in \Theta$, Proposition 3 provides a sharp pointwise approximation error bound as long as $Q_\lambda^{\pi_\theta} \in \mathcal{F}_{\bar{\nu}}$ with a corresponding transportation map $v_\theta$. In order for this result to hold, $v_\theta^\top(\theta_i(0))(s,a)\mathbb{1}_{\{\theta_i^\top(0)(s,a)\geq 0\}}$ is required to be iid for $i \in [m/2]$, which is the main idea behind the random initialization schemes for the NTK analysis. On the other hand, in policy optimization, $\theta(t)$ depends on the initialization $\theta(0)$, therefore $v_{\theta(t)}^\top(\theta_i(0))(s,a)\mathbb{1}_{\{\theta_i^\top(0)(s,a)\geq 0\}}$ is *not* independent – hence, $Cov(Y_i^\theta(s,a), Y_j^\theta(s,a)) \neq 0$ for $i \neq j$ in equation 67. Furthermore, the distribution of $(s,a)$ at time $t > 0$ also depends on $\pi_0$. Therefore, the pointwise approximation error cannot be used to provide an approximation bound for $Q_\lambda^{\pi_t}$ under the entropy-regularized NAC. In the existing works, this important issue regarding the temporal correlation and its impact on the NTK analysis was not addressed. In this work, we utilize the uniform approximation error bound provided in Proposition 3 to address this issue for $K$ functions to define the function class, while it may be possible to extend this to infinitely-many basis functions by using more general statistical complexity concepts.

In the absence of $Q_\lambda^{\pi_\theta}$, the critic yields a noisy estimate $\overline{Q}_\lambda^{\pi_\theta}$. Additionally, the samples $\{(s_n, a_n) \sim d_\mu^{\pi_\theta} \circ \pi_\theta(\cdot|s) : n \geq 0\}$ are used to obtain the update $u_t$. These two factors are the sources of error in the natural actor-critic method: $\min_u L_0(u, \theta(t)) \leq L_0(u_t, \theta(t))$. In the following, we quantify this error and show that:

1. Increasing number of SGD iterations, $N$,

2. Increasing representation power of the actor network in terms of the width $m$,

3. Low mean-squared Bellman error in the critic (by large $m', T'$),

lead to vanishing error.

First, we study the error introduced by using SGD for solving:

$$\widehat{L}_t(u, \theta(t)) = \mathbb{E}_{s \sim d_\mu^{\pi_t}, a \sim \pi_t(\cdot|s)}\left[\left(\nabla_\theta^\top \log \pi_t(a|s)u - \widehat{\Xi}_\lambda^{\pi_t}(s,a)\right)^2\right], \quad u \in \mathcal{B}_{m,R}^d(0), \tag{76}$$

Note that the auxiliary objective in equation 76 is Lipschitz continuous with modulus $q_{max} = 4(2R - \lambda\log\pi_{min})$ due to $\|u\|_2 \leq R$ for $u \in \mathcal{B}_{m,R}^d(0)$, $\|\nabla\log\pi_t(a|s)\| \leq 2\|\nabla f_t(s,a)\| \leq 2$ for ReLU networks, and the bound on $|\widehat{\Xi}_\lambda^{\pi_t}(s,a)| \leq 2|\overline{Q}_\lambda^{\pi_t}(s,a)| \leq 2|\overline{q}_{T'}^{\pi_t}(s,a) - \lambda\log\pi_t(a|s)| \leq 2(R - \lambda\log\pi_{min})$ that follows from $\pi_t(a|s) \geq \pi_{min}$ by Proposition 2 and $|\overline{q}_{T'}^{\pi_t}(s,a)| \leq R$ from the max-norm projection with radius $R$ for MN-NTD.

**Proposition 4** (Theorem 14.8, Shalev-Shwartz & Ben-David (2014))**.** *Algorithm 2 (Lines 2-2) with step-size* $\alpha_A = R/\sqrt{q_{max}^2 N}$ *yields the following result:*

$$\mathbb{E}[\widehat{L}_t(u_t, \theta(t))] - \min_u \widehat{L}_t(u, \theta(t)) \leq \frac{Rq_{max}}{\sqrt{N}}, \tag{77}$$

*for any* $R \geq \bar{\nu}$ *where the expectation is over the random samples* $\{(s_n, a_n) : n \in [N]\}$.

The following proposition provides an error bound in terms of the statistical error for finding the optimum $u_t$ via SGD as well as TD-learning error in estimating the soft Q-function. Let

$$L_t(u, \theta(t)) = \mathbb{E}_{s \sim d_\mu^{\pi_t}, a \sim \pi_t(\cdot|s)}\left[\left(\nabla_\theta^\top \log \pi_t(a|s)u - \Xi_\lambda^{\pi_t}(s, a)\right)^2\right]. \tag{78}$$

**Proposition 5.** *Let* $A = A_0 \cap A_1 \cap A_2$*, hence* $\mathbb{P}(A) \geq 1 - 3\delta$*. We have the following inequality:*

$$\mathbb{E}[\mathbb{1}_A L_t(u_t, \theta(t))] \leq 8 \min_u \left\{ \mathbb{1}_A L_0(u, \theta(t)) + \mathbb{E}\left[\mathbb{1}_A \big| [\nabla f_0(s,a) - \nabla f_t(s,a)]^\top u \big|^2\right] \right\} + \frac{2Rq_{max}}{\sqrt{N}}$$
$$+ 6\mathbb{E}[\mathbb{1}_A | Q_\lambda^{\pi_t}(s,a) - \overline{Q}_\lambda^{\pi_t}(s,a)|^2], \quad (79)$$

*where the expectation is over the samples for critic (TD learning) and actor (SGD) updates. Consequently, we have:*

$$\mathbb{E}[\sqrt{L_t(u_t, \theta(t))}] \leq 3\sqrt{\mathbb{E}[\min_{u \in \mathcal{B}_{m,R}^d(0)} L_0(u, \theta(t))]} + \frac{2\sqrt{Rq_{max}}}{N^{\frac{1}{4}}} + 3\rho_0(R/\lambda, m, \delta)$$
$$+ 3\mathbb{E}\sqrt{\mathbb{E}_{s,a}[|Q_\lambda^{\pi_t}(s,a) - \overline{Q}_\lambda^{\pi_t}(s,a)|^2]}, \quad (80)$$

*under the event A.*

*Proof.* We extensively use the inequality $(x + y) \leq 2x^2 + 2y^2$ for $x, y \in \mathbb{R}$. First, note that

$$\widehat{L}_t(u, \theta(t)) \leq 2L_t(u, \theta(t)) + 2\mathbb{E}_{s,a \sim d_t}[|Q_\lambda^{\pi_t}(s,a) - \overline{Q}_\lambda^{\pi_t}(s,a)|^2], \tag{81}$$

for any $u \in \mathcal{B}_{m,R}^d(0)$. Hence, under $A = A_0 \cap A_1 \cap A_2$, we have:

$$\mathbb{E}[L_t(u_t, \theta(t))] \leq 2\mathbb{E}[\widehat{L}_t(u_t, \theta(t))] + 2\mathbb{E}[|Q_\lambda^{\pi_t}(s,a) - \overline{Q}_\lambda^{\pi_t}(s,a)|^2], \tag{82}$$

$$\leq 2 \min_{u \in \mathcal{B}_{m,R}^d(0)} \widehat{L}_t(u, \theta(t)) + 2\mathbb{E}_{s,a}[|Q_\lambda^{\pi_t}(s,a) - \overline{Q}_\lambda^{\pi_t}(s,a)|^2] + \frac{2Rq_{max}}{N^{1/2}}, \tag{83}$$

$$\leq 4 \min_{u \in \mathcal{B}_{m,R}^d(0)} L_t(u, \theta(t)) + 6\mathbb{E}_{s,a}[|Q_\lambda^{\pi_t}(s,a) - \overline{Q}_\lambda^{\pi_t}(s,a)|^2] + \frac{2Rq_{max}}{N^{1/2}}, \tag{84}$$

where the second line follows from Prop. 4 and the last line follows from equation 81. Consequently, we have:

$$\mathbb{E}[L_t(u_t, \theta(t))] \leq 8 \min_{u \in \mathcal{B}_{m,R}^d(0)} \left\{ \mathbb{E}[(\nabla^\top f_0(s,a)u - Q_\lambda^{\pi_t}(s,a))^2] + \mathbb{E}[|(\nabla f_t(s,a) - \nabla f_0(s,a))^\top u|^2] \right\}$$
$$+ 6\mathbb{E}_{s,a}[|Q_\lambda^{\pi_t}(s,a) - \overline{Q}_\lambda^{\pi_t}(s,a)|^2] + \frac{2Rq_{max}}{N^{1/2}}, \quad (85)$$

where we use $(x + y)^2 \leq 2x^2 + 2y^2$ and the following inequality:

$$\mathbb{E}[(\nabla^\top \log \pi_t(a|s)u - \Xi_\lambda^{\pi_t}(s,a))^2] = \mathbb{E}_{s \sim d_\mu^{\pi_t}} Var(\nabla^\top f_t(s,a)u - Q_\lambda^{\pi_t}(s,a)),$$
$$\leq \mathbb{E}[(\nabla^\top f_t(s,a)u - Q_\lambda^{\pi_t}(s,a))^2].$$

Using equation 85 and Theorem 2 in Cayci et al. (2023) together with the inequality $\sqrt{x + y + z} \leq \sqrt{x} + \sqrt{y} + \sqrt{z}$ for $x, y, z > 0$, we obtain equation 80. $\qquad\square$

Hence, we obtain the following bound on the approximation error $\epsilon_{bias}^{\pi_t}$.

**Corollary 5** (Approximation Error). *Under Assumption 1-4, we have the following bound on the approximation error:*

$$\mathbb{E}[\mathbb{1}_A \cdot \epsilon_{bias}^{\pi_t}] \leq M_\infty \Big[ \frac{8\bar{\nu}}{\sqrt{m}} \Big( (d \log(m))^{\frac{1}{4}} + \sqrt{\log\Big(\frac{K}{\delta}\Big)} \Big) + \frac{2\sqrt{Rq_{max}}}{N^{\frac{1}{4}}} + 4\varepsilon \Big],$$

*where $A = A_0 \cap A_1 \cap A_2$, $m' = \widetilde{O}\Big( \frac{\bar{\nu}^4}{(1-\gamma)^2 \varepsilon^2} \Big)$ and $T' = O\Big( \frac{(1+2\bar{\nu})^2 \bar{\nu}^2}{\varepsilon^4} \Big)$ and $\mathbb{P}(A) \geq 1 - 3\delta$.*

*Proof.* In order to prove Corollary 5, we substitute the results of Corollary 4 and Lemma 2 into equation 80. □

The main message of Corollary 5 is as follows: in order to eliminate the bias introduced by using (i) function approximation, (ii) sample-based estimation for actor and critic, one should employ more representation power in both actor and critic networks (via $m$ and $m'$), and also use more samples in actor and critic updates (via $N$ and $T'$). Furthermore, Corollary 5 quantifies the required network widths and sample complexities to achieve a desired bias $\epsilon > 0$.

In the following subsection, we finally prove Theorem 1 by using the Lyapunov drift result (Lemma 6) and the approximation error bound (Corollary 5).

### A.5 Convergence of Entropy-Regularized Natural Actor-Critic

*Proof of Theorem 1.* In the following, we prove the first part of Theorem 1, where the second part follows identical steps with a constant step size $\eta \in (0, 1/\lambda)$. First, note that Lemma 6 implies the following bound:

$$\mathbb{1}_A \Big[ \Psi(\pi_{t+1}) - \Psi(\pi_t) \Big] \leq -\eta_t \lambda \Psi(\pi_t) \mathbb{1}_A - \eta_t (1-\gamma) \Delta_t \mathbb{1}_A + 2\eta_t^2 R^2 + \eta_t \mathbb{1}_A \epsilon_{bias}^{\pi_t}$$
$$+ (\eta_t \lambda + 6) \rho_0(R/\lambda, m, \delta) + 2\eta_t R \sqrt{\rho_0(R/\lambda, m, \delta)}. \tag{86}$$

By Corollary 5,

$$\mathbb{E}[\mathbb{1}_A \epsilon_{bias}^{\pi_t}] \leq M_\infty \Big[ \rho_1 + \frac{2\sqrt{Rq_{max}}}{N^{1/4}} + 4\varepsilon \Big] =: \epsilon_{bias},$$

where

$$\rho_1 = \frac{16\bar{\nu}}{\sqrt{m}} \Big( (d \log(m))^{\frac{1}{4}} + \sqrt{\log(K/\delta)} \Big).$$

Let $\overline{\Psi}_t := \mathbb{E}[\Psi(\pi_t) \mathbb{1}_A]$. Then,

$$\overline{\Psi}_{t+1} - \overline{\Psi}_t \leq -\eta_t \lambda \overline{\Psi}_t - \eta_t (1-\gamma) \mathbb{E}[\mathbb{1}_A \Delta_t] + 2\eta_t^2 R^2 + \eta_t \epsilon_{bias}$$
$$+ 7\rho_0(R/\lambda, m, \delta) + 2\eta_t R \sqrt{\rho_0(R/\lambda, m, \delta)}.$$

Since $\eta_t = \frac{1}{\lambda(t+1)}$, by induction,

$$\overline{\Psi}_{t+1} \leq (1 - \eta_t \lambda) \overline{\Psi}_t + \eta_t (1-\gamma) \mathbb{E}[\mathbb{1}_A \Delta_t] + 2\eta_t^2 R^2$$
$$+ \eta_t \Big( \epsilon_{bias} + 2R\sqrt{\rho_0(R/\lambda, m, \delta)} \Big) + 7\rho_0(R/\lambda, m, \delta),$$
$$\leq -\frac{(1-\gamma)}{\lambda(t+1)} \sum_{k \leq t} \mathbb{E}[\mathbb{1}_A \Delta_t] + \frac{1}{\lambda} \Big( \epsilon_{bias} + 2R\sqrt{\rho_0(R/\lambda, m, \delta)} \Big) + \frac{2R^2 \log(t+1)}{\lambda^2 (t+1)}$$
$$+ 4(t+1)\rho_0(R/\lambda, m, \delta).$$

Hence,

$$\min_{0 \leq t < T} \mathbb{E}[\mathbb{1}_A \Delta_t] \leq \frac{1}{1-\gamma} \Big( \epsilon_{bias} + 2R\sqrt{\rho_0(R/\lambda, m, \delta)} \Big) + \frac{2R^2(1 + \log T)}{\lambda(1-\gamma)T} + \frac{4T\lambda}{(1-\gamma)} \rho_0(R/\lambda, m, \delta), \tag{87}$$

which concludes the proof. □

*Proof of Corollary 2.* Note that, without the realizability assumptions, we have

$$\mathbb{E}\sqrt{\mathbb{E}[\min_{u \in \mathcal{B}_{m,R}^d(0)} L_0(u, \theta(t))]} \leq \epsilon_{\mathsf{app}}^{\mathsf{actor}},$$

and

$$3\mathbb{E}\sqrt{\mathbb{E}_{s,a}\big[|Q_\lambda^{\pi_t}(s,a) - \overline{Q}_\lambda^{\pi_t}(s,a)|^2\big]} \leq 3\sqrt{2}\left(\mathbb{E}[\sqrt{\mathbb{E}_{s,a}|Q_\lambda^{\pi_t}(s,a) - \check{Q}_\lambda^{\pi_t}(s,a)|^2} + \epsilon_{\mathsf{app}}^{\mathsf{critic}}\right),$$

where $\check{Q}_\lambda^{\pi_t}(s,a) \in \arg\min_{f \in \mathcal{F}_{\bar{\nu}}} \mathbb{E}_{s,a}(f(s,a) - Q_\lambda^{\pi_t}(s,a))^2$. Substituting these inequalities into equation 80 and replacing $\epsilon_{bias}$ accordingly, the proof follows from the same steps as the proof of Theorem 1. □

# B    Sampling from the Discounted State-Action Visitation Distribution

Given an initial state distribution $\mu$ and policy $\pi$, under a generative model that enables obtaining an independent and identically distributed $s^0$ from the initial state distribution $\mu$, the procedure summarized in Algorithm 4 yields an iid sample $(s,a)$ such that $s \sim d_\mu^\pi$ and $a \sim \pi(\cdot|s)$ (Konda & Tsitsiklis, 2003; Agarwal et al., 2020).

---

**Algorithm 4:** Sampler from $d_\mu^\pi \otimes \pi$ under a generative model

---

**inputs:** $\mu$: initial state distribution, $\pi$: policy, $\gamma$: discount factor;
Set $s_0 \sim \mu$ and $a_0 \sim \pi(\cdot|s_0)$;
**for** $k = 0, 1, \ldots$ **do**
    $i_k \sim \mathsf{Ber}(\gamma)$;
    **if** $i_k = 1$ **then**
        $s^{k+1} \sim P(\cdot|s^k, a^k)$;
        $a^{k+1} \sim \pi(\cdot|s^{k+1})$;
    **else**
        **return** $(s,a) = (s^k, a^k)$

---

The generative model enables the controller to access iid samples from the state-action visitation distribution $d_\mu^{\pi_t}$ by resetting from the initial state distribution $\mu$, which is critical in the actor-critic framework in Algorithm 2 that we analyzed in this paper. In the absence of such a generative model, the sampling procedure in an on-policy RL setting is performed by using a single Markovian trajectory (Sutton & Barto, 2018). Ergodicity is required in this case, which is always satisfied for irreducible and aperiodic Markov chains with a finite state space, where the verification of ergodicity for Markov chains with an infinite state space is more complicated (Norris, 1998). Under a Markovian sampling procedure in the ergodic setting, the distribution mismatch that led to the concentrability coefficient assumptions (Assumption 4') needs to be replaced by the stationary distribution $\{\xi^{\pi_t}\}_t$ under policies $\{\pi_t\}_t$ as

$$\max_{0 \leq t < T} \max_{(s,a) \in \mathcal{S} \times \mathcal{A}} \frac{d_\mu^{\pi^*}(s)\pi^*(a|s)}{\xi^{\pi_t}(s,a)} < \infty.$$

Furthermore, a finite-time analysis in the Markovian sampling would introduce new terms that depend on the mixing-times to analyze the transient behavior under each policy (Bhandari et al., 2021; Cayci et al., 2023).

## C   Proof of Lemma 7

Consider $g \in \mathcal{F}_{\bar{\nu}}$ with a corresponding transportation map $v \in \mathcal{H}_{\bar{\nu}}$. Using Cauchy-Schwarz inequality,

$$
\mathbb{E} \sup_{x:\|x\|_2 \leq 1} \left| g(x) - \frac{1}{m} \sum_{i=1}^{m} v^{\top}(\theta_i(0)) x \mathbb{1}\{\theta_i^{\top}(0) x \geq 0\} \right|^2
$$

$$
\leq \mathbb{E} \sup_{x:\|x\|_2 \leq 1} \left\| \frac{1}{m} \sum_{i=1}^{m} v(\theta_i(0)) \mathbb{1}_{\{\theta_i^{\top}(0) x \geq 0\}} - \mathbb{E}[v(\theta_i(0)) \mathbb{1}_{\{\theta_i^{\top}(0) x \geq 0\}}] \right\|^2.
$$

Define $b_i := v(\theta_i(0)) \mathbb{1}_{\{\theta_i^{\top}(0) x \geq 0\}}$. Define a class $B$ containing all possible values taken by $b := \{b_i\}_{i=1}^{m}$ over $\{x : \|x\|_2 \leq 1\}$ for a fixed $\theta(0)$. Further, using Cauchy-Schwarz inequality,

$$
\mathbb{E} \sup_{x:\|x\|_2 \leq 1} \left| g(x) - \frac{1}{m} \sum_{i=1}^{m} v^{\top}(\theta_i(0)) x \mathbb{1}_{\{\theta_i^{\top}(0) x \geq 0\}} \right|^2
$$

$$
\leq \mathbb{E} \sup_{b \in B} \frac{1}{m^2} \sum_{i \neq j}^{m} (b_i - \mathbb{E}b_i)^{\top} (b_j - \mathbb{E}b_j) + \frac{4\bar{\nu}^2}{m}.
$$

Using the symmetrization argument with Rademacher random variables $\sigma_{ij}$'s,

$$
\mathbb{E} \sup_{x:\|x\|_2 \leq 1} \left| g(x) - \frac{1}{m} \sum_{i=1}^{m} v^{\top}(\theta_i(0)) x \mathbb{1}_{\{\theta_i^{\top}(0) x \geq 0\}} \right|^2 \leq 4 \mathbb{E}_{\theta(0)} \mathbb{E}_r \sup_{b \in B} \frac{1}{m^2} \sum_{i \neq j}^{m} \sigma_{ij} b_i^{\top} b_j + \frac{4\bar{\nu}^2}{m}.
$$

Note that given $\theta(0)$, $B$ is a finite set. We apply Massart's Finite Class lemma to have,

$$
\mathbb{E}_r \left[ \sup_{b:b \in B} \frac{1}{m^2} \sum_{i \neq j}^{m} \sigma_{ij} b_i^{\top} b_j | \theta(0) \right] \leq \sqrt{\sum_{i \neq j}^{m} \|v(\theta_i(0))\|^2 \|v(\theta_j(0))\|^2} \frac{\sqrt{2 \log |B|}}{m^2}
$$

We calculate $|B|$ using VC-theory. Each element $b_i$ of $b$, partitions the space $\{\|x\| = 1\} \subset \mathbb{R}^d$ into two half planes where one half takes value $v(\theta_i(0))$ and another half takes value 0. Hence all possible values taken by $b$ in space $\{\|x\| = 1\}$ is equal to the number of components in the partition made by $m$ half planes $\{b_i\}_{i=1}^{m}$. The number of such components is bounded by $m^{d+1} + 1$ using the growth function defined by Vapnik & Chervonenkis (1971). Hence $|B| \leq m^{2d}$, and the following holds:

$$
\mathbb{E} \sup_{x:\|x\|_2 \leq 1} \left| g(x) - \frac{1}{m} \sum_{i=1}^{m} v^{\top}(\theta_i(0)) x \mathbb{1}_{\{\theta_i^{\top}(0) x \geq 0\}} \right|^2 \leq \frac{12 \bar{\nu}^2 \sqrt{d \log m}}{m}.
$$

The result follows from Jensen's inequality.

