# OpenReview forum: "Finite-Time Analysis of Entropy-Regularized Neural Natural Actor-Critic Algorithm"
_TMLR — Accepted by TMLR_

### Review · Reviewer_GATe · 2024-01-22

**Summary Of Contributions:**

The paper studies an entropy-regularized natural actor critic algorithm, utilizing two separate single-hidden-layer neural networks for the actor and critic, with a softmax parametrized policy. The paper provides a finite sample analysis of the proposed algorithms, and claims that it (1) provides a sample complexity result under mildest distribution mismatch conditions and a sharper sample complexity under a stronger distribution condition compared to prior work, and (2) shows that the combination of overparameterization, entropy regularization, gradient clipping, and averaging leads to the “persistence of excitation" condition, which then results in improved sample complexity.

**Audience:**

Yes

**Claims And Evidence:**

Yes

**Requested Changes:**

I suggest some changes to the paper:
- (critical) Justification for the assumptions and setting (see my comments in the previous part) and make sure all important details are covered in the paper.
- (nice to add) Discuss the novelty of the paper, highlighting new approaches or techniques used.
- (minor formatting fix) Citation style: please use \citet for citations within the text.

**Strengths And Weaknesses:**

**Strength:**
The paper provide a sound technical analysis on a popular (entropy-regularized) algorithm that might be interesting for some TMLR's audience. The claims appear correct, although the proofs in the appendix were not thoroughly verified.

**Weakness:**
The paper's main limitations are its clarity and might seem lack of novelty (although novelty isn't a primary criterion for TMLR evaluation):
- The paper misses some technical details, for example, why is the projection in (23) can be considered as gradient clipping?.
- The presentation order is sometimes confusing. For example, the authors mention MN-NTD in algorithm but MN-NTD is introduced in the next page. Another example is that the author mention that we want to control the max norm in Remark 1 but I guess the reason we want this is for Proposition 2? I think the order to present these can be improved.
- The paper isn't self-contained, often borrowing settings/assumptions/conditions from other works without adequate explanation. One example is the excitation condition (what is the original condition, what modification you made to make it appliable to your setting? What is the condition exactly?).
- Related to the previous comment, the paper does not provide sufficient justification for some assumptions and settings. For example, in section 3.1: “As a common practice, we fix the output layer c after a random initialization, and only train the weights of hidden layer”. It is totally reasonable, but the paper should still justify why it follows this practice.
- The current paper reads like a report that borrows tools from various sources, rather than a novel contribution. Elaboration on the new analysis techniques would enhance its novelty and impact. For example, the paper mention that “As a result, we have to use different analysis techniques”. Maybe the authors can follow up the sentence and explain potential questions such as what techniques are new?  Why previous work did not use these? Why did the authors use this particular condition?

In summary, I would suggest to accept the paper due to its technical merit and relevance. However, to improve its impact and clarity, I would suggest the authors to edit the paper.

Other question:
- In conclusion, the paper claims that “entropy regularization led to significantly improved sample complexity”, but I think the conclusion from the analysis is that we need overparameterization, entropy regularization, gradient clipping, and averaging?

---

> ### Author Response · Authors · 2024-02-18
>
> We would like to thank the reviewer for the very valuable and positive feedback, and for very constructive suggestions. In the revised manuscript, we have addressed the suggestions of the reviewer. Below we provide our detailed responses.
>
> * *Gradient clipping.* In the paper, the (natural) policy gradient $u_t$ is projected to ensure that $\|\|u_{i,t}\|\|_2\leq R/\sqrt{m}$ for each $i=1,2,\ldots,m$ given $R > 0$, such a norm-constraining operation on the gradient updates is commonly called gradient clipping. In the revised version, we clarify this point in Remark 1.
>
> * *Presentation.* In Section 3.1, we first provide the meta-algorithm to give an overview, and then we provide the details on the building blocks such as the critic (MN-NTD). Regarding the other comment, max-norm regularization is used to establish the so-called kernel regime, which takes place near the random initialization (Lemma 3), and also to prove sufficient exploration to ensure global optimality (Proposition 2), which are both critical to prove the main result Theorem 1. In the revised version, to address the suggestion of the reviewer, (i) we provide a clarification on these points in Section 3, and (ii) we provide additional pointers (e.g., in Remark 1 and Algorithm 2) to aid the reader navigate through the content seamlessly. We thank the reviewer for this suggestion.
>
> * *Persistence of excitation.* The persistence of excitation is a *general* condition that indicates a non-zero exploration probability of the decisions at any state, which is critical to ensure global convergence in a general stochastic control problem. Proposition 2 is an important contribution in our work, indicating that entropy-regularized Neural NAC ensures a strictly positive exploration probability $\pi_{min}>0$ at any state (i.e., persistence of excitation condition) with high probability over the random initialization for large network widths $m$, leading to the convergence result in Theorem 1 under minimal assumptions on the concentrability coefficients. We have clarified this in Section 4.1 to address the reviewer's suggestion.
>
> * *Training the output layer.* We have included an additional discussion in Section 4.1 on the impact of training the output layer, and the justification of training only the hidden layer. We have also provided additional discussions on other assumptions (e.g., realizability and sampling oracle assumptions).
>
> * *Contributions.* We have expanded our discussion on Remark 1 and the conclusion section to highlight our contributions. One important take-away message from our work is that, the combination of large network width $m$, entropy regularization, max-norm gradient clipping and weight-decay results in a positive exploration probability (Prop. 2), and (i) convergence under minimal assumptions on the concentrability coefficient thanks to Prop. 2, and (ii) significantly improved network width, sample complexity and iteration complexity bounds. We also point out important and unique challenges in the policy optimization framework from an approximation theory point of view, and establish new uniform approximation results to address these challenges, as indicated in the conclusion section and Section A.4.
>
> * *Algorithmic ideas that led to the improved bounds.* The reviewer is correct, the combination of entropy regularization, gradient clipping, weight decay and large network widths $m$ leads to the improved sample complexity results. We have corrected this in the revision.
>
> * *Citations.* We have revised the citation style with \citet{} as the reviewer suggested.

---

> ### Comment · Reviewer_GATe · 2024-02-20
> **Short response to the authors**
>
> Thank you for the response. I think the current paper is much clearer.
>
> I should clarify that I am not suggesting abandoning \citep entirely. I recommend using \citet for in-text citations like "X was introduced in (Konda & Tsitsiklis, 2000)." This should be rephrased as "X was introduced in Konda & Tsitsiklis (2000)." Another example, "(Kakade, 2001) proposed the natural gradient method" should be "Kakade (2001) proposed the natural gradient method". However, it is fine for sentences like "... a rich class of function approximation schemes (Mnih et al., 2016; Silver et al., 2016; Nachum et al., 2017; Duan et al., 2016)."

---

> ### Author Response · Authors · 2024-03-14
>
> We have revised the citation style as suggested by the reviewer. We would like to thank the reviewer very much again for providing very valuable and detailed feedback that helped us improve the overall quality of the paper.

---

### Review · Reviewer_8Z4o · 2024-01-31

**Summary Of Contributions:**

# Summary

The paper _Finite-Time Analysis of Entropy-Regularized Neural Natural Actor-Critic Algorithm_ presents a finite-time analysis of the natural actor-critic algorithm with neural network function approximation. The paper outlines how the natural actor-critic algorithm can attain provably good performance -- in terms of convergence, iteration complexity, sample complexity, _etc_. -- using entropy regularization and a number of implementation tricks including gradient clipping, weight decay, and averaging. The paper demonstrates that these methods are sufficient for avoiding policy collapse and convergence to sub-optimal policies. The paper also provides sample and iteration complexity bounds which improve on the existing bounds in the literature.

In particular, the paper results are proven for the natural actor-critic algorithm with single-hidden-layer neural network function approximation for both actor and critic.

**Audience:**

Yes

**Broader Impact Concerns:**

No broader impact concerns.

**Claims And Evidence:**

Yes

**Requested Changes:**

To secure my recommendation for acceptance, all issues presented in my **Main Argument** above would need to be addressed.

**Strengths And Weaknesses:**

# Main Argument

Although I am not an expert on policy gradient theory nor natural actor-critic algorithms, I believe that this paper provides ample theoretical proof to support the claims made therein. Compared to current works, the paper provides sharper sample and iteration complexity bounds for natural actor-critic algorithms. Further, the paper provides convergence guarantees in an entropy regularized setting with weaker assumptions than current works. In an entropy un-regularized setting, the paper improves upon current sample complexity bounds with standard distribution mismatch conditions. Finally, the theoretical analysis considers practices which are common in implementation, such as gradient clipping, weight decay, and entropy regularization -- things not often discussed in similar works. Nevertheless, the paper suffers from a number of issues which complicates the paper evaluation process:

In section 2, there are a number of errors. First, when discussing policies, the paper defines a policy as a function $\pi : \mathcal{S} \to \mathcal{A}$. Following, the paper states that a policy

> assigns some probability of taking an action $a \in \mathcal{A}$ at a given state $s \in \mathcal{S}$

These two definitions contradict each other. The first definition indicates that $\pi$ _deterministically_ selects an action in a given state (i.e. policy $\pi$ is a mapping from states to actions), whereas the second definition indicates that $\pi$ is a mapping from states to probability distributions over actions. Later, softmax distributions are used for policies, indicating that the first definition above is incorrect.


Next, in section 2.1, paragraph **Entropy-regularized objective**, the paper doubly defines _soft Q-functions_ as

\begin{equation}
q_\lambda^\pi(s, a) = \mathbb{E}_\pi \left[ \sum^\infty\_{k=0} \gamma^k (r(s_k, a_k) - \lambda \log \pi(a_k \mid s_k)) \mid s_0 = s, a_0 = a \right]
\end{equation}

and

$$
Q_\lambda^\pi(s, a) = r(s, a) + \gamma \mathbb{E}\_{s' \sim P(\cdot \mid s, a)} \left[ V\_\lambda^\pi (s') \right]
$$

These two functions are **not** equal -- there is a difference of $\lambda \log \pi(a \mid s)$ between the two. When the term _soft Q-function_ is used later, it is not clear which of these functions are referred to since both of these functions are defined as being the _soft Q-function_. Commonly, the second function is taken as the definition of the soft Q-function [1, 2, 3].


Next, in section 3.1, the optimization problem to estimate the natural gradient is defined as

\begin{equation}
u^*_t = \min\_{u \in \mathcal{B}^d\_{m, R}(0)} \mathbb{E} \left[ (\nabla^\top \log \pi_t (a \mid s) u - \Xi\_\lambda^{\pi_t}(s, a))^2 \right]
\end{equation}

This optimization problem resulted in a number of questions for me. First, what is the expectation with respect to? Judging based on Equation (15) from the paper, it appears that the expectation should be with respect to both $\pi_\theta$ and $d\_\mu^{\pi\_\theta}$. But, the paper indicates that this optimization problem is approximately solved _in the absence_ of $d\_\mu^{\pi_t}$ and $\Xi\_\lambda^{\pi_t}$, which makes me think that the expectation is not with respect to $d\_\mu^{\pi_t}$ at all. Later though, both in-text and in algorithm 2, the policy update uses samples $s_n \sim d\_\mu^{\pi_t}$ when solving this optimization problem. Are samples from $d\_\mu^{\pi_t}$ utilized or not? What does it mean to solve this optimization problem _in the absence of_ $d\_\mu^{\pi_t}$ when samples from $d\_\mu^{\pi_t}$ are used? Further, in this section, $\lambda$ is introduced as the _regularization parameter_, which is confusing since in preceding sections it was defined as the _entropy regularization scale parameter_. In section 3.1, is $\lambda$ the entropy regularization scale or is it a new parameter?

Next, the presented natural actor-critic algorithm seems to be theoretically justifiable, but many of the theoretical choices are not practically implementable. For example, in many real-world scenarios we cannot generate samples from $d\_\mu^{\pi_t}$, or else generating such samples is possible yet prohibitively expensive. Since no empirical justification is given for the algorithm, I am wondering if it would be possible to discuss such practical limitations in the paper, perhaps only in the appendix. For example, based on the convergence, sample complexity, and iteration complexity results, how large of a neural network might be usually needed in practice? Do the results extend to continuous action spaces? What ramifications exist if samples from $d\_\mu^{\pi_t}$ cannot be generated?

Next, the paper consistently makes the claim that the actor network learns a state-action value function. The actor network is outlined in equation (18):

\begin{equation}
f(s, a; (c, \theta)) = \frac{1}{\sqrt m} \sum\limits\_{i=1}^{m} c_i \sigma( \langle \theta, (s, a) \rangle)
\end{equation}

and the output of the actor networks is passed through a softmax function to generate a categorical policy distribution. To me, this looks like learning action preferences [7], and not action values. What exactly is meant in the paper by statements such as

> the uniform approximation power of the actor network to approximate Q-functions throughout policy optimization steps (pg 2)

How is the actor approximating a state-action value function?

Finally, some notation is confusing or undefined:
- In theorem 1 (and in multiple locations in the appendix), the notation $q\_{max}$ is undefined. What does it mean?
- The notations $\mathbb{1}\_{A}$, $\mathbb{1}\_{A_0 \cap A_1}$, $\mathbb{1}\_{A_2}$ _etc_. are used but left undefined. I believe this is the indicator function for the subscript argument. But, what exactly is the subscript? I could not find any indication in the main text of what the subscripts meant. I assume the subscripts refer to the corresponding equations in the appendix. This notation is used in Lemma 2, Theorem 1, Corollary 1, Corollary  2, and multiple locations in the appendix.
- Given the above point, it is confusing that $\mathbb{1}_A$ is the indicator function since the notation $\mathbb{1}${$A$} is used elsewhere, which also seems to be the indicator function for $A$. Are both of these notations referring to the indicator function? Can a single notation be used for the indicator function?


# Small Things

**These did not affect the scoring of this paper**

The **Neural network analysis** paragraph on page 3 has a discussion on first-order optimization methods for overparameterized neural networks. An important class of first-order optimization algorithms here are those related to mirror descent, which is closely related to natural gradient descent [5,6]. The current paper indicates that overparameterized neural networks trained using first-order optimization methods have a bias toward the minimum norm solution, but this phenomenon is actually a bit more general. Overparameterized neural networks trained using mirror descent optimization methods have a bias toward the minimum Bregman divergence (in mirror map) solution [4].

A number of grammatical errors exist in the paper:

- "On of the main goals of..." -> "On**e** of the main goals of..."
- Many times, references are used as nouns. For example "PG methods were established in (Agarwal et al., 2020)." Instead, this would be better written as "Agarwal et al. (2020) established PG methods".
- In section 1.1, it is noted that the _policy optimization is stabilized by incorporating ... averaging_, but it is not clear from context what is being averaged.


# References

[1] Tuomas Haarnoja, Aurick Zhou, Kristian Hartikainen, George Tucker, Sehoon Ha, Jie Tan, Vikash Kumar, Henry Zhu, Abhishek Gupta, Pieter Abbeel, Sergey Levine. Soft Actor-Critic Algorithms and Applications. 2019.

[2] Tuomas Haarnoja, Aurick Zhou, Pieter Abbeel, Sergey Levine. Soft Actor-Critic: Off-Policy Maximum Entropy Deep Reinforcement Learning with a Stochastic Actor. ICML. 2018.

[3] Matthieu Geist, Bruno Scherrer, Olivier Pietquin. A Theory of Regularized Markov Decision Processes. ICML. 2019.

[4] Navid Azizan, Sahin Lale, Babak Hassibi. Stochastic Mirror Descent on Overparameterized Nonlinear Models: Convergence, Implicit Regularization, and Generalization. IEEE Transactions on Neural Networks and Learning Systems. 2022.

[5] Garvesh Raskutti, Sayan Mukherjee. The Information Geometry of Mirror Descent. IEEE Transactions on Information Theory. 2015.

[6] Suriya Gunasekar, Blake Woodworth, Nathan Srebro. Mirrorless Mirror Descent: A Natural Derivation of Mirror Descent. Proceedings of The 24th International Conference on Artificial Intelligence and Statistics. 2021.

[7] Richard Sutton and Andrew G. Barto. Reinforcement Learning: An Introduction. 2nd Edition. MIT Press. 2018.

---

> ### Author Response · Authors · 2024-02-18
>
> We would like to thank the reviewer for the very valuable and positive feedback, and for very constructive suggestions. In the revised manuscript, we have addressed the suggestions of the reviewer. Below we provide our detailed responses.
>
> **Corrections**
>
> We thank the reviewer for reading our paper in detail, and pointing out the typos and other errors. Below we provide the list of changes we have made in this revision to address the suggestions by the reviewer.
>
> * *Policy definition.* We have clarified that the action $a$ is chosen at state $s$ with probability $\pi(a|s)$ under a policy $\pi$.
> * *Q-function and soft Q-function.* We have corrected the name for $q_\lambda^\pi$ as the (entropy-regularized) Q-function. $Q_\lambda^\pi$ is called the soft Q-function under a policy $\pi$.
> * *Indicator function $\mathbb{1}$.* $\mathbb{1}_A$ refers to the indicator function of an event $A$. We have now unified the notation for $\mathbb{1}$ throughout the paper, and added a definition in Section 1.3.
> * Definition of $q_{max}$: We have now provided the definition of $q_{max}$ in Theorem 1, and provided a discussion on it after equation (76). It is the modulus of Lipschitz continuity for the compatible function approximation error in equation (76), which yields the natural policy gradient update.
> * We have corrected the typos pointed out by the reviewer. Also, we have revised the citation styles according to the TMLR formatting instructions, as suggested by the reviewer.
>
>
> **Sampling from a state visitation distribution**
> * *Expectation in the definition of* $u_t^*$. The expectation in the definition of $u_t^*$ in Section 3.1 is with respect to $s\sim d_\mu^{\pi_t}$ and $a\sim\pi_t(\cdot|s)$. We have now clarified this in the revised version of the paper.
> * *Sampling from* $d_\mu^\pi$. We use samples from the state visitation distribution $d_\mu^{\pi_t}$. This distribution $d_\mu^{\pi_t}$ is **not** known a priori by the controller as it requires the knowledge of the transition kernel $P$, but getting samples from this unknown distribution $d_\mu^\pi$ is possible in a simple way without knowing or learning $P$. In the revision, Algorithm 4 describes this sampling process from $d_\mu^\pi$.
> * *On the sampling procedure.* Our paper assumes the availability of a sampling oracle that provides iid samples from the state-visitation distribution, to simplify the already-complicated analysis. In applications without such a generative model, a single trajectory under Markovian sampling is used. This different sampling strategy brings new challenges, such as a different definition of the concentrability coefficient, and mixing-time analyses to address the transient behavior of the Markov chains throughout the policy optimization steps. As suggested by the reviewer, we have now included a discussion on this in Section B in the appendix regarding the necessary changes under different sampling procedures. We also point out the discussion in Section 3.2 on sampling for the critic.
> * *Regularization parameter.* Throughout the paper, the variable $\lambda > 0$ denotes the regularization parameter. We have clarified this in the revision.
>
> **Approximation by the actor network**
>
> In the revision, we have included Corollary 2, which explicitly shows the approximation errors that stem from the use of actor and critic networks. In the actor approximation error in (38), the term $\nabla^\top f_0(s,a) u$ is the first-order approximation of the actor network $f(s, a; (c, u+\theta(0)))$ with an error bounded by $O(1/\sqrt{m})$ as we establish in Lemma 3 in the kernel regime. Thus, the representation power of the actor network in approximating $Q_\lambda^{\pi_t},~t=0,1,\ldots$, which is characterized by $\epsilon_\mathsf{app}^\mathsf{actor}$, directly impacts the optimality gap as in Corollary 2. The discussion in Section A.4 provided a finer characterization of the approximation error for the actor, presented in Corollary 2, for our main result in Theorem 1.

---

> > ### Comment · Reviewer_8Z4o · 2024-03-04
> > **Response**
> >
> > I would like to thank the authors for their response and for the alterations made which have improved the clarity of the paper and theoretical results.

---

> > > ### Author Response · Authors · 2024-03-14
> > >
> > > We would like to thank the reviewer very much again for providing very valuable and detailed feedback that helped us improve the overall quality of the paper.

---

### Review · Reviewer_Qxav · 2024-02-08

**Summary Of Contributions:**

This paper considers an entropy-regularized natural actor-critic algorithm with neural network function approximation, and provides a finite-time analysis. Specifically, they provide sharp bounds on sample complexity, convergence rate, and network width. They introduce this notion of "persistence of excitation" to suggest that the combination of over-parametrization, entropy regularization, gradient clipping, and averaging leads to sufficient exploration without the need to assume this as done in previous analyses.

**Audience:**

Yes

**Broader Impact Concerns:**

No concerns.

**Claims And Evidence:**

Yes

**Requested Changes:**

I would like more discussion/treatment around the weaknesses raised above included in the main text. A minor note, Equations 5 and 7 were both referred to as a "soft Q-function", despite being different (as acknowledged in the text)- this may cause confusion and should be emphasized.

That said, these are points that would simply strengthen the work in my view, and I am overall positive about the work in its current state.

**Strengths And Weaknesses:**

Overall, as far as I could check, the analyses are correct. I particularly found the persistence of excitation discussion interesting as it validated existing techniques. My main concerns lie in 1) the strength and reasonableness of the assumptions made, and 2) what can be taken away from the work? To elaborate:

**Strengths**:
1. The paper is nicely structured and written.
2. They characterize trade-offs in the analyzed algorithm, and provide some understanding around existing techniques in the literature. While it's intuitive that entropy-regularization has implications for exploration, it was interesting to see the projection radius and network width's role in it characterized.
3. They provide sharp bounds on sample complexity, iteration complexity, and network width. As far as I was able to check, the analyses appear correct.

**Weaknesses**:
1. Assumptions 2 and 3 on realizability seem strong- particularly Assumption 3. Can the authors elaborate on the necessity of these assumptions, and whether or not they can be weakened? Is there any analyses or empirical evidence validating that such inherent structure can be reasonably expected, i.e., are they realistic?
2. The work is purely theoretical, largely contributing characterizations and understanding of an algorithm. However, it's unclear if more can be concluded from the results as the algorithm and the specific modifications to it are relatively widespread already. To clarify, I see value in the theoretical contribution presented, and in the analysis techniques adopted, but are there any concrete take-aways for someone looking to use some form of entropy-regularized neural actor-critic on some problem? e.g., the network width bounds and discussion around the critic only needing to approximate the value for a single policy at a given moment vs the actor needing to approximate the policies corresponding with the sequence of critics (Remark 5), could be emphasized as some practical consideration in the conclusion.

---

> ### Author Response · Authors · 2024-02-18
>
> We would like to thank the reviewer for the very valuable and positive feedback, and for very constructive suggestions. In the revised manuscript, we have addressed the suggestions of the reviewer. Below we provide our detailed responses.
>
> **Realizability Assumptions and the Approximation Error**
>
> We would like to thank the reviewer for this important question. In order to address these questions and comments on Assumptions 2-3, in this revision,
> * we have expanded the discussion on the realizability assumptions in Section 4.4 and 4.2 in addition to Section A.4,
> * we have also included a new result in Corollary 2, which provides a finite-time bound *without* the realizability assumptions by including the previously-omitted function approximation errors that stem from the use of neural networks.
>
> *Convergence result without realizability assumptions.* We can eliminate the realizability assumptions, and this relaxation will bring approximation errors, which depend on the approximation capabilities of the neural networks used in the actor and the critic. These error terms, $\epsilon_\mathsf{app}^\mathsf{critic}=\max_t\mathbb{E}\min_{f\in\mathcal{F}^{\bar{\nu}}}\sqrt{\mathbb{E}(f(s,a)-q_\lambda^{\pi_t}(s,a))^2}$ and $\epsilon_\mathsf{app}^\mathsf{actor}=\max_t\mathbb{E}\min_{u\in\mathcal{B}}\sqrt{\mathbb{E}(\nabla^\top f_0(s,a)u-Q_\lambda^{\pi_t}(s,a))^2}$, appear in the finite-time error bound in Corollary 2 as $O(\frac{M_\infty}{1-\gamma}(\epsilon_\mathsf{app}^\mathsf{critic}+\epsilon_\mathsf{app}^\mathsf{critic}))$, omitting the realizability assumptions.
>
> *Discussion on the realizability assumptions.* The strategy for approximation error analysis, which would yield a finer characterization of the terms $\epsilon_\mathsf{app}^\mathsf{critic},\epsilon_\mathsf{app}^\mathsf{critic}$ in the kernel regime, is to establish universal approximation results in the infinite-width limit, and bounding the projection error between the infinite-width limit and its projection onto the class of finite-width neural networks, which is usually $O(1/\sqrt{m})$ (Proposition 3).
>
> For the critic, the function class in Approximation 2 is a subclass of the reproducing kernel Hilbert space (RKHS) associated with the neural tangent kernel, including functions whose RKHS norm is upper bounded by $\bar{\nu}$. Thus, since the NTK is a universal kernel, $\mathcal{F}_{\bar{\nu}}$ can approximate any continuous functions on a compact set (as is the case in our paper) arbitrarily well (i.e., with arbitrarily small critic approximation error) for sufficiently large $\bar{\nu}$ by the universal approximation result for the NTK [Ji et al., 2019], justifying Assumption 2.
>
> For the actor, as the reviewer noted, the approximation error has significantly different characteristics than that of the critic, since the actor network (same once initialized) has to approximate all soft Q-functions throughout the policy optimization steps $t=0,1,\ldots,T-1$ for near-optimality by Equation (62). This is a critical difference that our paper reveals between the policy optimization problem with dynamically changing target functions and the supervised learning or policy evaluation problem with a static target (labeling) function.
>
> As we argue in Section A.4, a fine characterization of the actor approximation error is significantly challenging. In Section A.4, we go one step further than Corollary 2, and characterize the class of policies $\pi_\theta$ whose soft Q-functions lie in a rich function class that can be approximated well by a finite-width neural network by establishing uniform approximation bounds (Proposition 3). Assumption 3 indicates the soft-Q functions should lie in the convex hull of $K$ functions in $\mathcal{F}_{\bar{\nu}}$. For sufficiently large $\bar{\nu}$ and $K$, this class of functions can approximate the rich class of *separable* subspaces of continuous functions, as argued in Section 4.2. Since $m$ scales with only $O(\log{K})$ in our results, we believe that this yields a reasonable assumption from a bias-complexity perspective.
>
> Finally, in this work, we considered a uniform approximation result based on the use of finitely many basis functions for $\mathcal{F}_{K,\bar{\nu},\mathcal{V}}$, while it may be possible to establish more complicated approximation results by using general statistical complexity results to incorporate infinitely many basis functions. We discussed this in the conclusion section.
>
> **On the contributions and main take-aways**
>
> We totally agree with the reviewer about emphasizing the main take-aways, as one important objective in our work is to identify which algorithmic ideas yield provably optimal performance in this framework. To that end, we have highlighted the main take-aways from our work in the conclusion section, as suggested by the reviewer.
>
> **Typos**
>
> We thank the reviewer for pointing out the typo. We corrected $q_\lambda^\pi$ as the (entropy-regularized) Q-function.

---

> > ### Comment · Reviewer_Qxav · 2024-03-05
> > **Response**
> >
> > I appreciate the very thorough response and the resulting revisions from the authors, especially the explicit characterization of the approximation errors without realizability assumptions! The overall presentation and clarity has improved and my concerns have been satisfactorily addressed. :)

---

> > > ### Author Response · Authors · 2024-03-14
> > >
> > > We would like to thank the reviewer very much again for providing very valuable and detailed feedback that helped us improve the quality of the paper. We are very glad that we have addressed the concerns of the reviewer.

---

### Decision · Action_Editor_Pigr · 2024-03-14

**Recommendation:** Accept as is

**Comment:**

Generally the reviewers were positive on the paper and had detailed reviews, many questions and suggestions for clarifications. The authors responded in detail.

**Audience:**

Relevant for RL researchers in general, but particularly those working on actor-critics.

**Claims And Evidence:**

The work contributes a finite-time analysis for an entropy-regularized natural actor-critic algorithm with neural network function approximation. All three reviewers, all with significant experience in the area, agreed on acceptance, noting no issues with the analysis. As one would hope the analysis sheds light on current practice and offers suggestions for improved implementations.